# Dual therapy with corticosteroid ablates the beneficial effect of DP2 antagonism in chronic experimental asthma

Md Ashik Ullah[1], Sonja Rittchen[1,2], Jia Li[1,3], Bodie F. Curren [1,4],
Patricia Namubiru[1,4], Tufael Ahmed[1], Daniel R. Howard[1,4],
Muhammed Mahfuzur Rahman[1,4], Md Al Amin Sikder[1,4], Ridwan B. Rashid[1,4],
Natasha Collinson[1], Mary Lor[1], Mark L. Smythe[5] & Simon Phipps [1,4,6,7] ✉

Prostaglandin D2 (PGD2) signals via the DP1 and DP2 receptors. In Phase II trials, DP2 antagonism decreased airway inflammation and airway smooth muscle (ASM) area in moderate-to-severe asthma patients. However, in Phase III, DP2 antagonism failed to lower the rate of exacerbations, and DP2 as a target was shelved. Here, using a preclinical model of chronic experimental asthma, we demonstrate that rhinovirus-induced exacerbations increase PGD2 release, mucus production, transforming growth factor (TGF)-β1 and type-2 inflammation. DP2 antagonism or DP1 agonism ablates these phenotypes, increases epithelial EGF expression and decreases ASM area via increased IFN-γ. In contrast, dual DP1-DP2 antagonism or dual corticosteroid/DP2 antagonism, which attenuates endogenous PGD2, prevented ASM resolution. We demonstrate that DP2 antagonism resolves ASM remodelling via PGD2/DP1-mediated upregulation of IFN-γ expression, and that dual DP2 antagonism/corticosteroid therapy, as often occurred in the human trials, impairs the efficacy of DP2 antagonism by suppressing endogenous PGD2 and IFN-γ production.

Acute exacerbation of chronic asthma, most commonly as a result of rhinovirus (RV) infection, causes significant morbidity, mortality and healthcare costs[1]. A critical pathological feature of chronic asthma is airway remodeling characterized by goblet cell hyperplasia, airway smooth muscle (ASM) hyperplasia, and reticular basement membrane thickening. In an acute exacerbation of chronic asthma, the remodeled airways undergo bronchoconstriction causing airway narrowing and air-trapping leading to increased wheezing and breathlessness[2]. Commonly used treatments, including corticosteroids or non-steroidal anti-inflammatory drugs, fail to halt or reverse ASM remodeling[3], while the emerging monoclonal antibody therapies have

not been assessed sufficiently[4], and hence the prevailing view remains that ASM remodeling, once established, is irreversible. Although bronchial thermoplasty reduces ASM mass, this procedure is only recommended for individuals with uncontrolled-severe asthma, which limits its therapeutic application[5]. Thus, there is an unmet need for new therapeutics to reverse established airway remodeling.

In addition to the type-2 'instructive cytokines' thymic stromal lymphopoietin, IL-33, and IL-4, the eicosanoid prostaglandin D2 (PGD2) has been shown to amplify type-2 inflammation by activating multiple effector cells, including CD4+ T helper 2 ($T_H2$) cells, type 2 innate lymphoid cells (ILC2), eosinophils, mast cells, and basophils[6].

[1]Respiratory Immunology Laboratory, QIMR Berghofer Medical Research Institute, Herston, QLD 4006, Australia. [2]Otto Loewi Research Center for Vascular Biology, Immunology and Inflammation, Division of Pharmacology, Medical University of Graz, Graz 8010, Austria. [3]Department of Laboratory Medicine, Shanghai General Hospital, Shanghai Jiao Tong University School of Medicine, Shanghai 20080, China. [4]School of Biomedical Sciences, University of Queensland, Brisbane, QLD 4072, Australia. [5]Institute for Molecular Bioscience, University of Queensland, Brisbane, QLD 4072, Australia. [6]Australian Infectious Diseases Research Centre, University of Queensland, Brisbane, QLD 4072, Australia. [7]School of Biomedical Sciences, Queensland University of Technology, Brisbane, QLD 4000, Australia. ✉e-mail: simon.phipps@qimrberghofer.edu.au

Additionally, PGD2 directly activates ASM cells and is a potent bronchoconstrictor[7]. These effects are mediated via activation of the PGD2 receptor 2 (DP2), and this PGD2-DP2 pathway is associated with severe, poorly controlled, type-2-high asthma[8]. Consequently, several DP2 antagonists were developed for the treatment of asthma, and in phase-II clinical trials, DP2 blockade was shown to decrease airway eosinophilia, improve lung function and delay recurrence of an asthma exacerbation in patients with mild-to-moderate asthma or inadequately controlled, severe asthma[9–15]. Intriguingly, in patients with persistent eosinophilic asthma, DP2 antagonism was observed to lower ASM area[11], although the mechanism of action remains unknown. Despite these promising findings, large-scale phase III trials in patients with uncontrolled or severe asthma found that the DP2 antagonist fevipiprant only modestly (and not significantly) decreased the rate of exacerbations compared with placebo, and failed to improve lung function[16,17]. Similarly, in a Phase IIb study, GB001 failed to significantly reduce the odds of asthma worsening[18]. Precisely why the DP2 antagonists were ineffective in these latter trials remains unclear, as does the mechanism of action by which fevipiprant successfully reverses ASM remodeling.

In a pre-clinical model of viral bronchiolitis and using primary human bronchial epithelial cells, we previously demonstrated that the beneficial effects of DP2 antagonism, including the attenuation of type-2 inflammation and the restoration of antiviral immunity, are largely mediated via the activation of the PGD2 receptor DP1[19], indicating that the beneficial effects of DP2 antagonism are dependent on the production of endogenous PGD2. As corticosteroids inhibit and downregulate the expression of cyclooxygenase (COX)2 and subsequent PGD2 production[20–22], we hypothesized that the continuation of standard-of-care corticosteroid treatment, as occurred in the majority of trials, may have decreased the effectiveness of DP2 antagonism in the treatment of asthma. Here, using an established high-fidelity pre-clinical mouse model of chronic experimental asthma (CEA), characterized by eosinophilic inflammation and persistent ASM remodeling[23–25], we sought to test this hypothesis and to investigate the molecular mechanism by which DP2 antagonism reverses ASM remodeling. We demonstrate that DP2 antagonism, by favouring PGD2/DP1 signaling, induces the production of IFN-γ. As a consequence, TGF-β1 expression decreases and epithelial epidermal growth factor (EGF) expression increases, leading to a reduction in ASM area. As this phenotype is dependent on the production of endogenous PGD2, which is suppressed by corticosteroids, the beneficial effects of DP2 antagonism are ablated by dual therapy.

## Results

### Treatment with a DP2 antagonist, but not a corticosteroid, ameliorates airway remodeling during a RV-triggered exacerbation of chronic experimental asthma

To investigate the immune mechanisms by which DP2 antagonism reverses established airway remodeling, we employed a high-fidelity mouse model of CEA previously established in our laboratory[23,24]. Inoculation of WT mice from infancy with the respiratory virus, pneumonia virus of mice (PVM; a close relative of human respiratory syncytial virus) and cockroach extract (CRE) (Fig. 1A), leads to the development of CEA, as demonstrated by persistent airway smooth muscle (ASM) remodeling and collagen deposition[23,24]. The model simulates the human epidemiology linking severe/frequent lower respiratory infections and allergic sensitization that synergistically increase the risk of childhood asthma[26]. To mimic an acute asthma exacerbation, mice were inoculated with RV four weeks later, and immunopathology was assessed at 1, 3, and 7 days post-infection (dpi). Consistent with our previous findings and that of others[19,27,28], respiratory viral infection led to a significant increase in PGD2 levels in the lung compared to non-infected control mice (Fig. 1B), suggesting that PGD2 contributes to viral-triggered exacerbations of asthma.

Treatment with a DP2 antagonist (timapiprant, also known as OC000459) or a DP1 agonist (BW245c) prior and during the RV infection (Fig. 1A) had no effect on lung PGD2 levels at 1 or 7 dpi, although DP2 antagonism increased PGD2 levels at 3 dpi (Fig. 1B). Consistent with our previous findings, ASM area and peri-bronchial collagen deposition were significantly elevated in mice with CEA compared to naïve controls (Fig. 1C, D, F and Supplementary Fig. S1A)[25]. Neither pathology was further increased following RV exposure; however, treatment with the DP2 antagonist during RV infection significantly decreased ASM area, mirroring the findings of the phase II clinical trial of fevipiprant[11] and decreased peri-bronchial collagen (Fig. 1C, D, F). A similar outcome was observed following treatment with a DP1 agonist (Fig. 1C, D, F). We compared this treatment effect to fluticasone, a mainstay treatment for asthma, and to soluble (s)IL-13Rα2, a treatment that neutralizes IL-13, a central mediator of type-2 inflammation. Consistent with clinical investigations[29,30], fluticasone had little effect on ASM area or collagen deposition (Fig. 1E, G). sIL-13Rα2 similarly had no effect ASM area but did lower collagen deposition (Fig. 1E, G). RV inoculation increased mucus production at 3 and 7 dpi (Fig. 1H), and this effect was attenuated by DP2 antagonism or DP1 agonism as well as by fluticasone or sIL-13Rα2 treatment (Fig. 1H, I). Collectively, these data indicate that DP2 antagonism or DP1 agonism during a virus-induced exacerbation of asthma resolves established airway remodeling, including the increased ASM mass, whereas corticosteroid treatment only prevents the increase in mucus production.

### DP2 antagonism or DP1 agonism decreases lung TGF-β1 expression

Transforming growth factor β1 (TGF-β1) contributes to ASM remodeling and collagen synthesis and is elevated in the lungs of patients with asthma[31,32]. Here, we observed increased TGF-β1 immunoreactivity around the airways of mice with CEA compared to their naïve counterparts (Fig. 2A, B). Following RV inoculation, TGF-β1 expression was further increased around the airways, a phenotype confirmed by measuring TGF-β1 (by ELISA) in whole lung homogenates (Fig. 2A, B, D). Significantly, DP2 antagonism or DP1 agonism ablated this increase in TGF-β1 expression and lowered TGF-β1 levels to those observed in naïve controls (Fig. 2A, B, D). In contrast, neither fluticasone nor sIL-13Rα2 treatment affected TGF-β1 expression (Fig. 2C, E). These data suggest that DP2 antagonism or DP1 agonism resolves airway remodeling by suppressing the expression of TGF-β1. We next evaluated whether other growth factors (amphiregulin; epidermal growth factor, EGF), innate cytokines (IL-1β; IL-6), or epithelial alarmins (IL-33; IL-25; thymic stromal lymphopoietin) associated with allergic inflammation and tissue remodeling were markedly affected by both DP2 antagonism and DP1 agonism. Of these, the only mediator whose expression was (i) significantly affected by both treatments at >1 time point, and (ii) either higher or lower across all three treatment time points, was EGF (Fig. 2F and Supplementary Fig. S1B–D). The phenotype observed by ELISA was largely replicated when EGF immunoreactivity was quantified in the airway epithelium (Fig. 2H, I). EGF immunoreactivity was absent after preincubating the antibody with exogenous EGF, confirming specificity (Supplementary Fig. S1E). Notably, in contrast to TGF-β1, EGF expression negatively correlated with ASM area, but as shown for TGF-β1, EGF expression was unaffected by fluticasone or sIL-13Rα2 treatment (Fig. 2G, J).

### DP2 antagonist-mediated resolution of airway remodeling is dependent on endogenous PGD2 and ablated by dual therapy with a corticosteroid

The effectiveness of the DP1 agonist in resolving established ASM remodeling suggested that DP2 antagonism primarily elicits its beneficial effects via DP1 receptor activation, and thus requires endogenous PGD2. To test this, we hypothesized that the beneficial effects of DP2

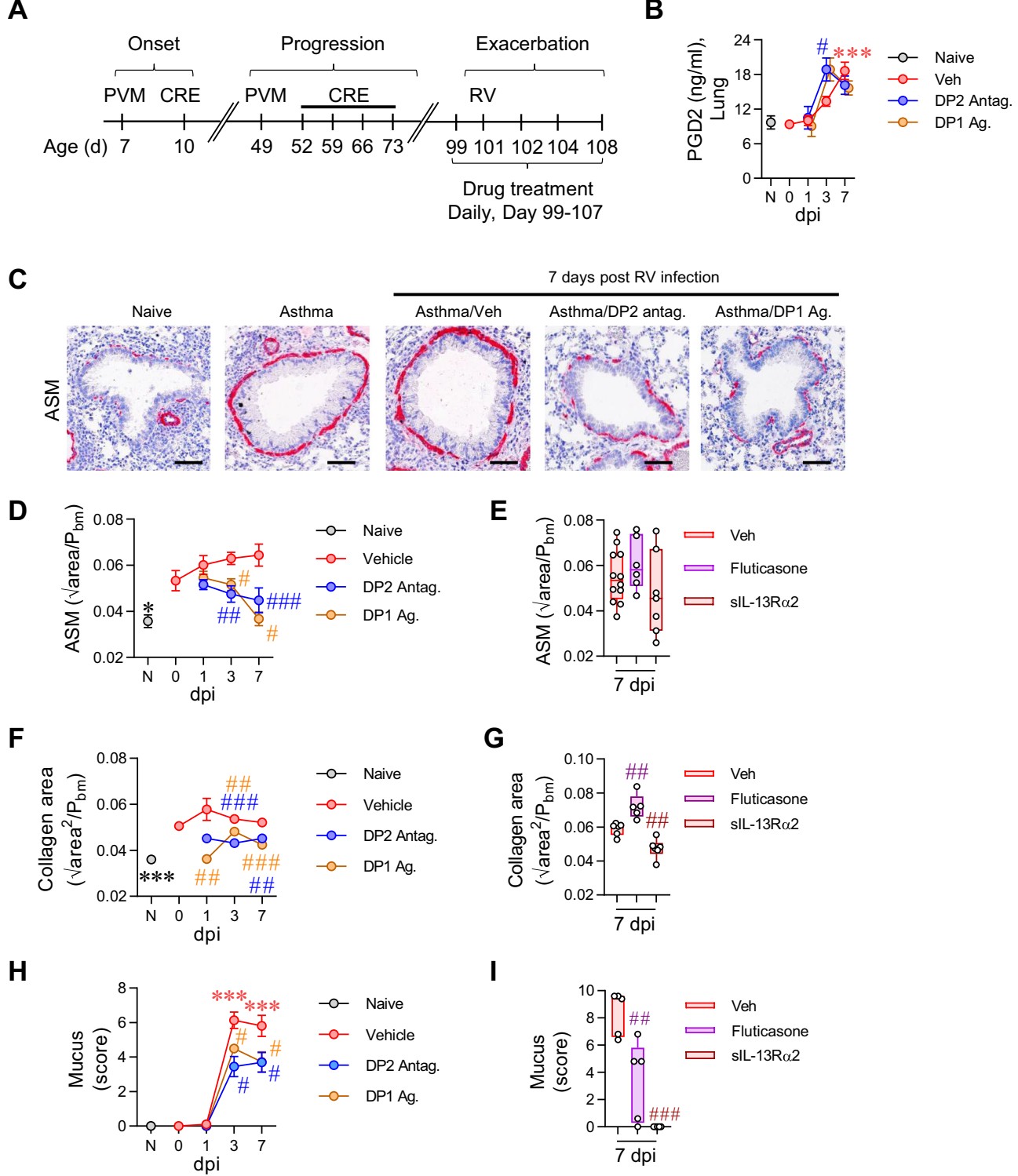

**Fig. 1 | Treatment with a DP2 antagonist, but not a corticosteroid, ameliorates airway remodeling during a RV-triggered exacerbation of CEA. A** Study design. Seven days old mice were inoculated with PVM and exposed to CRE 3 days later. Mice were re-infected with PVM six weeks later and exposed to CRE weekly for four weeks. Four weeks later, mice were inoculated with RV-1b. Separate groups of mice were treated with a DP2 antagonist (OC000459), DP1 agonist (BW245c), flutica-sone, soluble IL-13Rα2 or vehicle. Mice exposed to vehicle instead of virus or allergen were referred to as naïve. Mice were then inoculated with RV-1b and euthanized at 1, 3 and 7 days post infection (dpi). **B** Lung PGD2 levels. **C** Representative lung histology of α-smooth muscle actin (SMA) expression. Scale bars = 50 μm. **D**, **E** Airway smooth muscle (ASM) area. **F**, **G** Collagen deposition. **H**, **I** Mucus score. Data are presented as mean ± SEM or box-and-whisker plots showing individual data points with the boxes representing quartiles and whiskers indicating the range and are pooled data from two independent experiments ($n = 4–16$ mice per group). Statistical significance between different time points or different groups was determined using one-way ANOVA with Dunnett's multiple comparison test. * denotes $p < 0.05$, ** denotes $p < 0.01$ and *** denotes $p < 0.001$ compared to Vehicle group. # denotes $p < 0.05$, ## denotes $p < 0.01$ and ### denotes $p < 0.001$ compared to RV-infected group at corresponding time point. Source data are provided as a Source Data file.

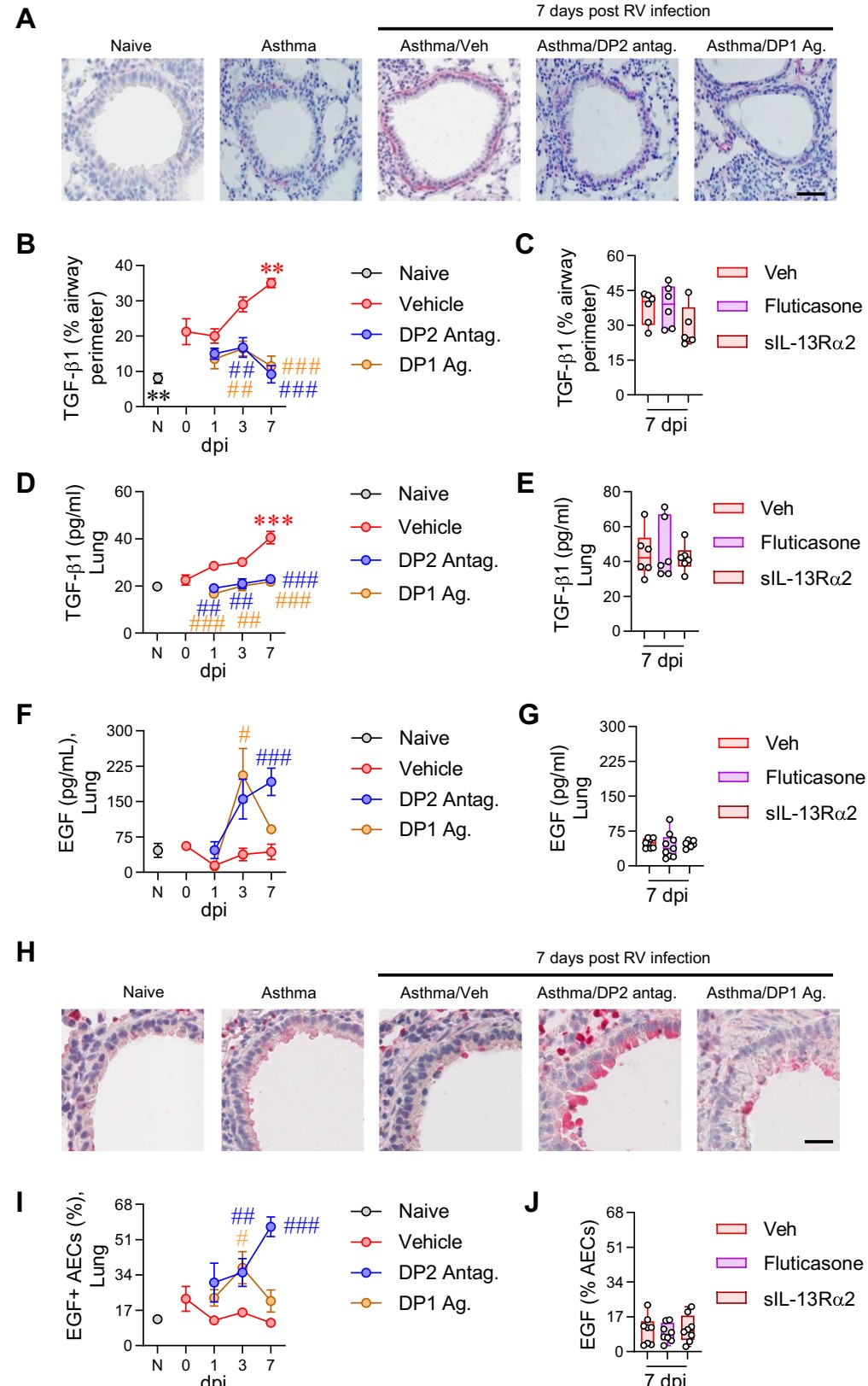

antagonism (i) would not occur in the absence of RV-induced inflammation, (ii) would be ablated by dual DP1/DP2 antagonism, and (iii) would not be replicated by inhibition of hematopoietic prostaglandin D synthase (h-PGDS), since this lowers endogenous PGD2, ablating the activation of both DP1 and DP2. As predicted, treatment with a DP2 antagonist in the absence of a RV-induced exacerbation (akin to 'stable' asthma) failed to affect ASM area, collagen deposition, TGF-β1 or EGF expression (Supplementary Fig. S2A–F). Notably, in RV-inoculated mice with CEA, dual antagonism with a DP2 antagonist and a DP1-specific antagonist (MK0524) prevented the resolution of airway remodeling and attenuated the fall in TGF-β and increase in EGF expression mediated by DP2 antagonism (Fig. 3A–E). The dual DP1/DP2

**Fig. 2 | DP2 antagonism or DP1 agonism decreases lung TGF-β1 expression and increases EGF expression. A** TGF-β1 immunoreactivity (red coloration). Scale bars = 50 μm. **B, C** Quantification of TGF-β1 immunoreactivity around the airways. **D, E** Lung TGF-β1 levels in homogenates. **F, G** Lung EGF levels in homogenates. **H** EGF immunoreactivity (red coloration). Scale bars = 50 μm. **I, J** Quantification of EGF immunoreactivity in airway epithelial cells (AECs). Data are presented as mean ± SEM or box-and-whisker plots showing individual data points with the boxes representing quartiles and whiskers indicating the range and are pooled data from two independent experiments ($n = 4$–$9$ mice per group). Statistical significance between different time points or different groups was determined using one-way ANOVA with Dunnett's multiple comparison test. ** denotes $p < 0.01$ and *** denotes $p < 0.001$ compared to vehicle group. ## denotes $p < 0.01$ and ### denotes $p < 0.001$ compared to RV-infected group at corresponding time point. Source data are provided as a Source Data file.

antagonism lowered PGD2 production (Fig. 3F), consistent with the notion that PGD2/DP1 activation acts in a positive feed-forward loop, which presumably enhances its efficacy. Despite decreasing PGD2 levels, h-PGDS inhibition failed to affect ASM area, collagen deposition, mucus hypersecretion, EGF or TGF-β1 expression (Supplementary Fig. S3). The requirement for endogenous PGD2 suggested that the effectiveness of DP2 antagonism would be decreased by co-treatment with a corticosteroid, as corticosteroids inhibit phospholipase A2, and lower the expression of cyclooxygenases and h-PGDS, thereby decreasing prostanoid production[20–22]. Significantly, dual fluticasone/DP2 antagonism treatment ablated the beneficial effect of DP2 antagonism (Fig. 3G–K), an effect associated with a significant decrease in lung PGD2 levels (Fig. 3L).

### DP2 antagonism or DP1 agonism attenuates RV-induced type-2 inflammation and promotes type-1 immunity
Next, to explore the mechanism by which DP1 agonism reverses airway remodeling, we investigated the effect of DP2 antagonism or DP1 agonism on RV-induced airway inflammation (gating strategies in Supplementary Fig. S4). In response to RV, neutrophil numbers increased sharply at 1 dpi and then waned by 3 dpi, whereas eosinophil numbers rose steadily and peaked at 7 dpi (Fig. 4A). Treatment with the DP2 antagonist or DP1 agonist decreased the infiltration of both granulocytes at 1 and 3 dpi, although at 7 dpi, both interventions led to an increase in lung neutrophils (Fig. 4A, B). Similar to the temporal pattern of eosinophilia, GATA3+ $T_H2$ cells and ILC2 numbers tended to increase between 3 and 7 dpi, and both cell types were decreased in response to DP2 antagonism or DP1 agonism (Fig. 4B). The type-2 effector cytokines, IL-4, IL-5, and IL-13 were elevated, particularly at 1 and 3 dpi, and significantly decreased by either treatment (Fig. 4C and Supplementary Fig. S5A). Similarly, IL-17A expression, which was elevated at 1 dpi, was ablated by DP2 antagonism or DP1 agonism at 1 dpi (Fig. 4D) and aligned with a reduction in lung RORγt+ $T_H17$ cells but not ILC3s (Supplementary Fig. S5B, C). In contrast, IFN−γ expression in mice with CEA was numerically (but not statistically) lower than that of naïve controls at baseline and decreased further at 3 and 7 dpi (Fig. 4E). The pattern of IFN−γ expression was not associated with T-bet+ $T_H1$ cells, ILC1s, or NK cells (Supplementary Fig. S5B, C). Following treatment with the DP2 antagonist or DP1 agonist, the expression of IFN−γ and TNF, another type-1 cytokine, was significantly increased (Fig. 4E). Of note, dual DP1/DP2 antagonist or dual fluticasone/DP2 antagonist treatment abrogated the increase in IFN−γ and TNF expression induced by DP2 antagonist therapy alone (Fig. 4F, G). Dual fluticasone/DP2 antagonist treatment also diminished the suppressive effect of DP2 antagonism monotherapy on IL-4 and IL-5 levels, although IL-13 levels remained significantly lower than the vehicle-treated mice (Supplementary Fig. S5D). Collectively, these data demonstrate that (i) DP2 antagonism or DP1 agonism attenuates type-2 inflammation and promotes type-1 immunity and (ii) dual DP2 antagonism/corticosteroid therapy prevents this shift to type-1 immunity.

### DP2 antagonism or DP1 agonism enhances NK cell IFN−γ production and alters the phenotype of alveolar macrophages
To identify the cell types producing IFN−γ and TNF at 7 dpi, we performed intracellular cytokine staining. RV inoculation decreased the number of IFN−γ producing cells, and this fall was arrested by DP2 antagonist or DP1 agonist treatment (Fig. 5A). Several cell types expressed IFN−γ, however the predominant sources were CD4+ T cells, CD8+ T cells and NK cells (Fig. 5B and Supplementary Fig. S6A), with only the number of IFN−γ-producing NK cells being increased by both treatments. TNF-producing cells increased in number following RV infection and were further increased by either intervention (Fig. 5C and Supplementary Fig. S6B). The predominant cell types expressing TNF were CD4+ T cells, type-2 classical dendritic cells (cDC2) and alveolar macrophages (AM); however, only TNF-producing AM numbers were increased by both treatments (Fig. 5D, E). Consistent with other reports[33], viral inoculation significantly lowered lung AM numbers at 1 and 3 dpi. However, by 7 dpi, the AM population had recovered and was significantly elevated compared to the naïve controls (Fig. 5F). As IFN−γ is known to contribute to the generation of CD86+CD206−/low TNF-producing AMs[34], we next evaluated the effect of DP2 antagonism or DP1 agonism on the numbers of CD86highD206−/low (M1-like macrophages) and CD86−/lowCD206high (M2-like macrophages) AMs during the RV exacerbation. As expected, CD86−/lowCD206high AMs were significantly elevated in mice with CEA (Fig. 5G). Although neither intervention affected the total numbers of AMs (Fig. 5F), treatment with either the DP2 antagonist or DP1 agonist attenuated the increase in CD86−/lowCD206high AMs between 3 and 7 dpi (Fig. 5G), and increased CD86highCD206−/low AM numbers (Fig. 5G). A similar pattern was apparent when interstitial macrophages and monocytes were analyzed, with both populations showing fewer CD86−/lowCD206high AMs, particularly at 3 dpi (Supplementary Fig. S6C, D). Collectively, these data demonstrate that treatment with a DP2 antagonist or DP1 agonist during an RV-induced exacerbation increases IFN−γ−producing NK and T cell numbers, which is associated with a shift in the AM phenotype towards a CD86HighCD206− M1-like phenotype that produces TNF.

### DP2 antagonism mediates airway remodeling resolution via enhanced IFN−γ production
To investigate whether IFN−γ contributes to the resolution of airway remodeling in response to DP2 antagonism, mice were treated with anti-IFN−γ and the DP2 antagonist during an RV-triggered exacerbation (Supplementary Fig. S7A). Anti-IFN−γ ablated the increase in IFN−γ levels induced by DP2 antagonism (Supplementary Fig. S7B), counteracted the resolving effects of DP2 antagonism on airway remodeling (Fig. 6A–C) and ablated the decrease in TGF-β1 expression (Fig. 6D and Supplementary Fig. S7C) and increase in EGF expression (Fig. 6E and Supplementary Fig. S7D). Neutralization of IFN−γ reversed the suppressive effect of DP2 antagonism on type-2 inflammation and lowered PGD2 production (Supplementary Fig. S7E–H). Consistent with a role for IFN−γ in inducing CD86+ AMs, anti-IFN−γ attenuated the DP2 antagonist-mediated shift from CD206+ AMs to CD86+ AMs, and prevented the associated increase in TNF levels (Fig. 6F–H). Taken together, these data suggest that the beneficial effects of DP2 antagonism on airway remodeling depend on the production of IFN−γ, which suppresses type-2 inflammation and the production of TGF-β1.

### Discussion
In a mouse model of CEA, we demonstrated that DP2 antagonism during a RV-triggered exacerbation lowers established ASM mass and collagen deposition, mirroring the observation in the Phase II trial with fevipiprant. This phenotype was replicated by a DP1 agonist, and the

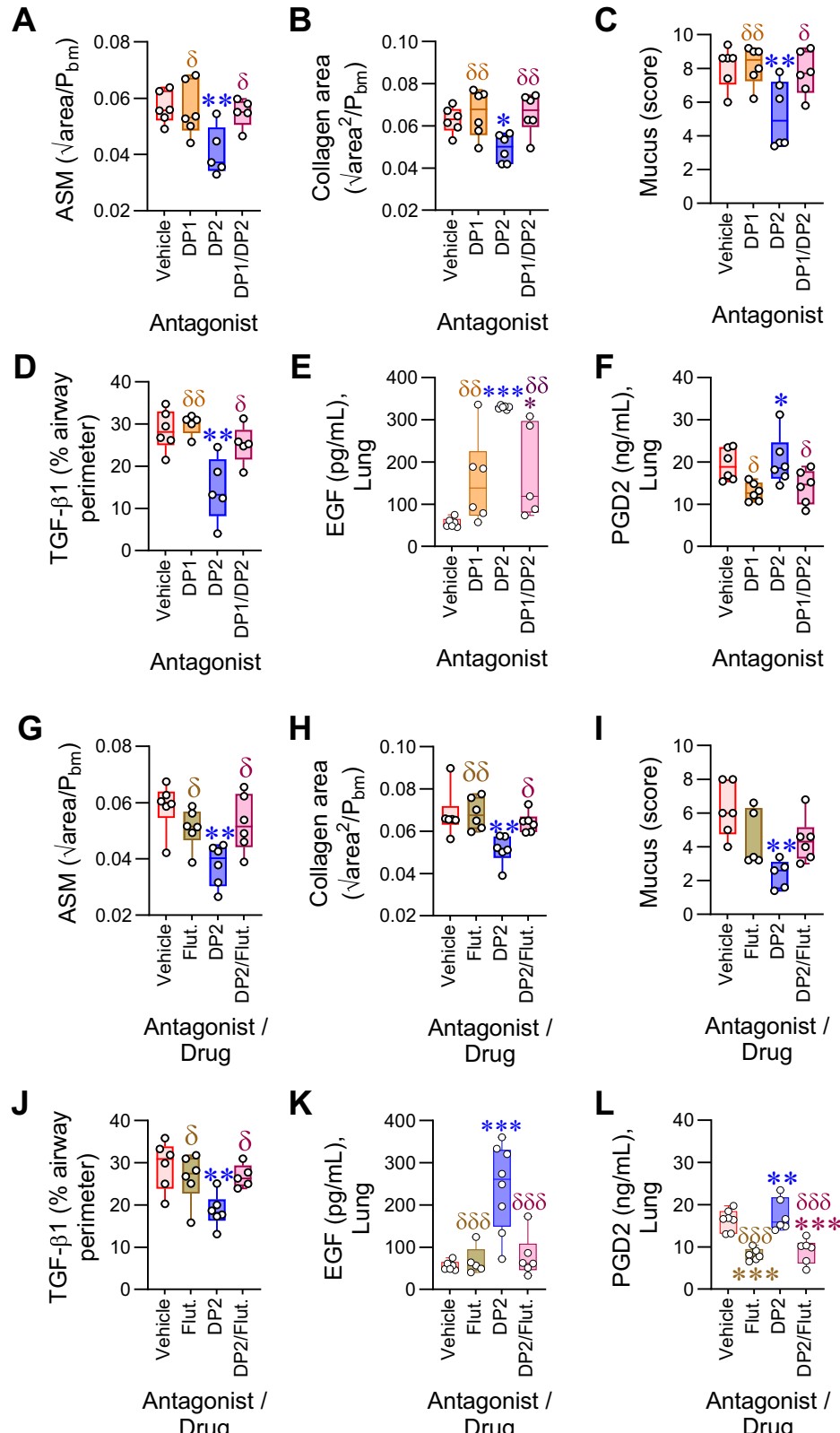

beneficial effect of DP2 antagonism was ablated by DP1 antagonism, implicating the critical and beneficial role of endogenous PGD2-mediated activation of DP1. Corticosteroids dampened the RV-induced increase in endogenous PGD2 levels necessary for the resolution of ASM remodeling, and hence dual steroid/DP2-antagonist therapy impeded the effectiveness of DP2 antagonism. Mechanistically, we identified that DP2 antagonism lowers the expression of TGF-β1, a

known ASM mitogen and pro-fibrotic cytokine, and increases epithelial EGF expression, and that these effects were mediated by increased levels of IFN-γ (Fig. 7).

In many of the Phase II trials of DP antagonists (fevipiprant, setipiprant, timapiprant), encouraging outcomes were observed[9–15]. However, in the Phase III trials, fevipiprant led to only a modest reduction in the rates of asthma exacerbations in patients with

**Fig. 3 | DP2-antagonist mediated resolution of airway remodeling is dependent on endogenous PGD2 and ablated by dual therapy with a corticosteroid.**
**A–E** Mice with CEA were Inoculated with RV-1b and given vehicle, DP2 antagonist or DP1 antagonist or both DP2 and DP1 antagonist daily from day 99 and euthanized at 7 dpi. Quantification of (**A**) ASM, (**B**) collagen deposition, (**C**) Mucus score and (**D**) TGF-β1 expression. Lung (**E**) EGF and (**F**) PGD2 levels in homogenates. **G–L** Mice with chronic asthma were inoculated with RV-1b and treated with vehicle, DP2 antagonist, fluticasone, or both drugs daily from day 99 and euthanized at 7 dpi. Quantification of (**G**) ASM, (**H**) collagen deposition, (**I**) Mucus score and (**J**) TGF-β1

expression. Lung (**K**) EGF and (**L**) PGD2 levels in homogenates. Data are presented as box-and-whisker plots showing individual data points with the boxes representing quartiles and whiskers indicating the range and are representative of two independent experiments ($n = 5$–7 mice per group). Statistical significance between different groups was determined using one-way ANOVA with Dunnett's multiple comparison test. * denotes $p < 0.05$, ** denotes $p < 0.01$ and *** denotes $p < 0.001$ compared to vehicle group. δ denotes $p < 0.05$ and δδ denotes $p < 0.01$ compared to DP2 antagonist treated group. Source data are provided as a Source Data file.

inadequately controlled severe asthma, and thus failed to meet the primary efficacy endpoint[16,17]. The underlying reasons for these disappointing results remain unclear, although it's important to recognize the design and endpoint of the Phase III trials were completely different to the Phase II trials, and the lower than expected number of exacerbations in the placebo arm decreased the window for improvement in the Phase III trials[16]. Nevertheless, the 22% reduction in exacerbations in the patients that received the 450 mg dose suggests that a sub-group of patients were responsive[16]. Careful inspection of the data from the phase II trial shows that although a significant reduction in ASM area was observed, this phenotype was not observed in all patients, with some showing no benefit and 2 of 14 patients an increase in ASM area[11]. The factors that underlie this response heterogeneity are varied, and would include the various endophenotypes that underlie chronic asthma, gene-environment interactions activated across the trial duration, and medication use, compliance and efficacy (e.g. steroid responsiveness). Our study highlights that the effectiveness of DP2 antagonism is critically dependent on the production of endogenous PGD2, and thus the inhibitory effects of steroids on phospholipase A2 and COX2/PGDS expression[20–22] may have counteracted the positive effects of DP2 antagonism in some patients, potentially explaining the mixed success of the various Phase II and III trials[9–18,35]. Our findings revealed that DP2 antagonism in the absence of a RV-induced exacerbation has no benefit and that dual DP2 antagonism/steroid therapy is ineffective, suggesting that immune cells actively contribute and are necessary for the beneficial effects of DP2 antagonism. In a Phase II study that employed an experimental RV challenge of patients with asthma, DP2 antagonism did not affect exacerbation severity and of note, the investigators reported a negative correlation between PGD2 levels during RV infection and prescribed ICS dose[35], consistent with the notion that steroids decrease the effectiveness of DP2 antagonism. Paradoxically, these observations suggest that therapy with a DP2 antagonist would work best in those patients who frequently exacerbate (given the need for endogenous PGD2), who are poorly compliant with their ICS/OCS therapy, or whose steroid insensitivity/resistance fails to dampen the production of PGD2. Since our findings highlight a key role for endogenous PGD2 for the efficacy of DP2 antagonists, future studies should consider the identification and targeting of the most suitable study population, perhaps through measuring nasal PGD2 production at baseline and in response to a viral mimetic, or following the stimulation of peripheral blood mononuclear cells ex vivo. Another approach to improve efficacy would be to combine a DP2 antagonist with a DP1 agonist or a drug that induces endogenous PGD2, such as a TLR7 ligand, to elicit a more favourale outcome.

Consistent with the findings from the Phase II trial in asthma and our previous publication in the context of viral bronchiolitis[11,19], we observed that DP2 antagonism or DP1 agonism reversed established ASM area, mucus hypersecretion, and collagen deposition. Of note, we identified that inoculation with RV increased TGF-β1 expression in the ASM bundles, and that DP2 antagonism not only prevented this effect, but markedly decreased TGF-β1 expression by ASM cells. In contrast, the expression of epithelial EGF was inversely correlated with ASM area and TGF-β1, suggesting that it contributes to ASM resolution.

Increased EGF receptor expression has long been associated with asthma severity[36,37], and is perceived as evidence of an injury and repair response since ligation of the receptor by EGF promotes reepithelialisation and wound healing[37,38]. In general, EGF levels are reported as equivalent or lower in individuals with asthma or allergic rhinitis compared to healthy controls[36,39,40]. However, amphiregulin, heparin binding EGF, and TGF-α also signal via the EGF receptor, and more recently, oxidized IL-33 was shown to activate EGFR when complexed with RAGE (receptor for advanced glycation endproducts), and in this context, it is noteworthy that EGF receptor signaling inhibits wound healing[37,41–44]. Our findings suggest that the DP2 antagonist- or DP1 agonist-mediated shift in the microenvironment of EGF receptor-active cytokines (e.g. higher EGF, higher amphiregulin, and lower IL-33) favors the restitution of normal physiology in the airway mucosa; that is, the resolution of goblet cell and ASM hyperplasia. Notably, in the phase II trial, in addition to decreasing ASM area, active treatment with fevipiprant led to increased epithelial integrity[11]. Consistent with previous reports[19], we observed that DP2 antagonism downregulated type-2 inflammation and critically, induced a reciprocal upregulation of type-1 cytokines (IFN-γ and TNF). Moreover, we identified that the increase in IFN-γ expression mediated the pro-resolving effect of DP2 antagonism on ASM remodeling. Fluticasone lowered type-2 inflammation but did not affect ASM area or collagen deposition. In line with the broad-spectrum anti-inflammatory effects of corticosteroids, fluticasone also ablated the increase in IFN-γ and TNF expression induced by DP2 antagonism, and accordingly had no effect on TGF-β1 or EGF expression, explaining the lack of effect on ASM area. Interestingly, NK cells were the only IFN-γ-producing cell type whose numbers were increased by both DP2 antagonism or DP1 agonism, although a limitation of our study is that we did not directly explore the role of this cell in mediating the protective effects of DP2 antagonism. The precise mechanism of action by which IFN-γ promoted the resolution of airway remodeling also remains an open question, although we demonstrated that anti-IFN-γ ablated the respective increase and decrease in EGF and TGF-β1 levels induced by DP2 antagonism. As AECs and ASM cells express the IFN−γ receptor, IFN-γ may mediate its ASM-resolving effects directly (e.g. by decreasing TGF-β1) or indirectly (e.g. by increasing EGF), or both. Additionally, IFN-γ can promote the differentiation of CD86[high]CD206[-/low] AMs[34,45] which, in direct contrast to CD86[-/low]CD206[high] AMs, inhibit fibrosis by releasing anti-fibrogenic factors like TNF[46–48]. In our study, DP2 antagonism increased CD86[high]CD206[-/low] AMs and decreased CD86[-/low]CD206[high] AMs, and elevated TNF levels, and these phenotypes, all associated with a reduction in ASM area, were reversed following IFN-γ blockade. Further studies are needed to unravel the specific contribution of NK cells and AMs in mediating the resolution of airway remodeling. Although a defined feature of asthma, we did not assess airway hyperreactivity, or lung function, similar to the phase III trials, where a severe exacerbation was graded according to systemic corticosteroid use for 3 days or more and an admission to hospital or an emergency department visit. Rather our study focused on elucidating novel cellular and molecular insights since such information will provide new tractable targets for future interventions, as has occurred for mediators such as thymic stromal lymphopoietin.

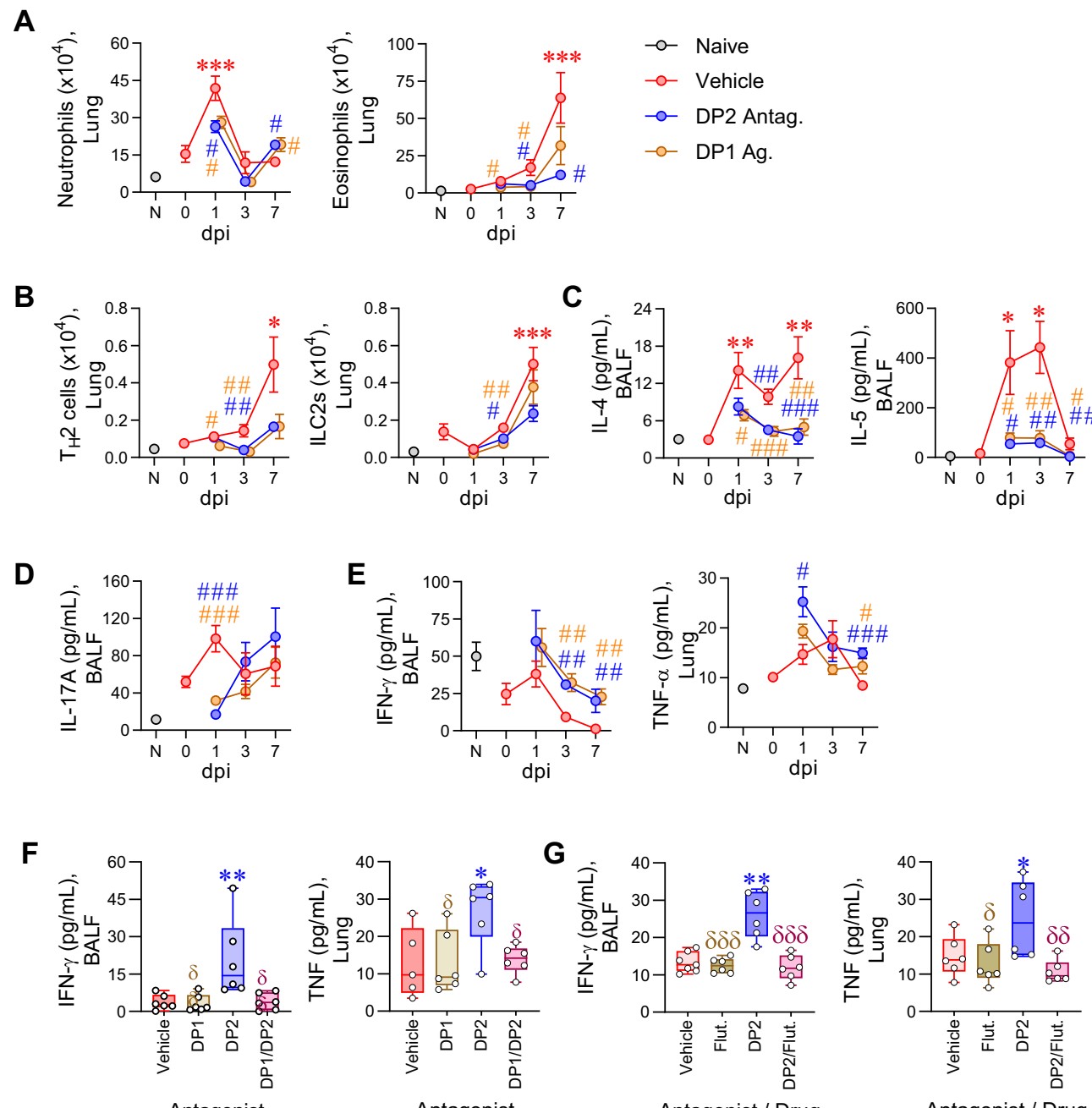

**Fig. 4 | DP2 antagonism or DP1 agonism attenuates RV-induced type-2 inflammation and promotes type-1 immunity.** Mice with CEA were inoculated with RV-1b and treated daily with a DP2 antagonist, a DP1 agonist, or vehicle starting from day 99. Mice were euthanized at 1, 3 and 7 dpi. **A** Number of neutrophils (SigF⁻CD11b⁺Ly6G⁺CD11c⁻) and eosinophils (SigF⁺CD11b⁺Ly6G⁻CD11c⁻) in the lungs. **B** Number of $T_H2$ cells (GATA3⁺CD4⁺T) and ILC2s (CD3ε⁻CD19⁻CD45R⁻CD11c⁻Gr-1⁻CD11b⁻NK1.1⁻ CD90.2⁺CD200R1⁺ GATA3⁺) in the lungs. **C** IL-4 and IL-5 concentration in the bronchoalveolar lavage fluid (BALF). **D** IL-17A expression in the BALF. **E** IFN-γ expression in the BALF and TNF expression in the lung. **A**–**E** Data are presented as mean ± SEM and are representative of two independent experiments showing similar results ($n = 4$–11 mice per group). Statistical significance between different time points or different groups was determined using one-way ANOVA with Dunnett's multiple comparison test. * denotes $p < 0.05$; ** denotes $p < 0.01$ and *** denotes $p < 0.001$ compared to Vehicle group. # denotes $p < 0.05$, ## denotes

$p < 0.01$ and ### denotes $p < 0.001$ compared to RV-infected group at corresponding time point. **F** RV-1b infected mice were treated with vehicle or DP2 antagonist (OC000459) or DP1 antagonist (MK0524) or both DP2 and DP1 antagonist daily from day 99 and euthanized at 7 dpi. Concentrations of IFN-γ in the BALF and TNF in the lung. **G** RV-1b infected mice were treated with vehicle, DP2 antagonist, fluticasone, or both DP2 antagonist and fluticasone daily from day 99 and euthanized at 7 dpi. Concentrations of IFN-γ in the BALF and TNF in the lung. **F**, **G** Data are presented as box-and-whisker plots showing individual data points with the boxes representing quartiles and whiskers indicating the range and are representative of two independent experiments showing similar results ($n = 5$–7 mice per group). * denotes $p < 0.05$ and ** denotes $p < 0.01$ compared to vehicle group. δ denotes $p < 0.05$, δδ denotes $p < 0.01$ and δδδ denotes $p < 0.001$ compared to DP2 antagonist treated group. Source data are provided as a Source Data file.

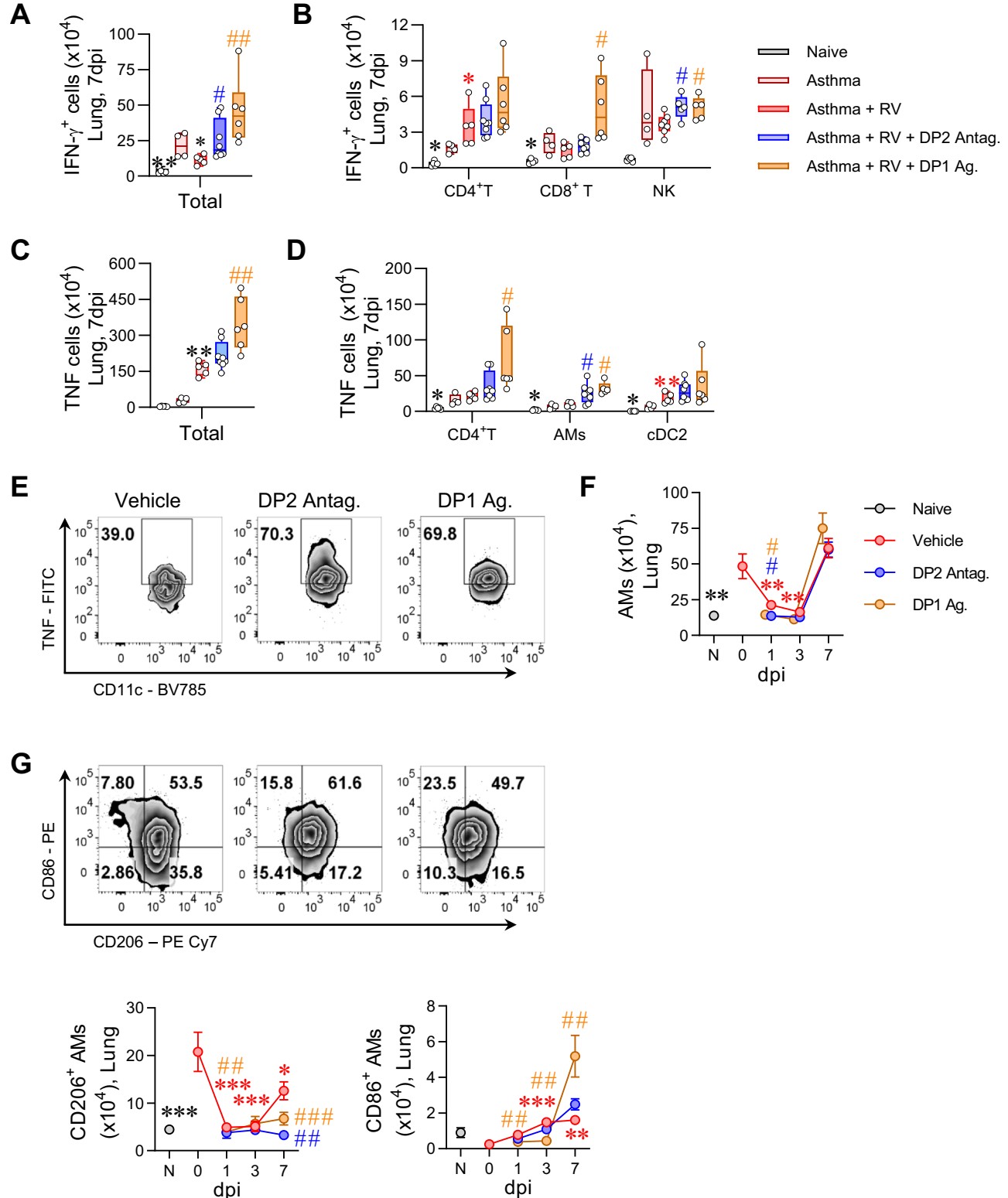

In conclusion, using a high-fidelity preclinical model of CEA, we found that DP2 antagonism during a RV-induced asthma exacerbation resolves established ASM remodeling. Mechanistically, the beneficial effect of DP2 antagonism was mediated by increased levels of IFN-γ, which decreased TGF-β1 expression by ASM cells and increased EGF expression by airway epithelial cells. Notably, the therapeutic efficacy of DP2 antagonism was dependent on endogenous PGD2-mediated activation of DP1, and ablated by dual therapy with a corticosteroid.

The use of DP2 antagonists for the treatment of other indications that are characterized by small airway fibrosis and ASM remodeling, and for which steroids are not the mainstay treatment, such as chronic obstructive pulmonary disease, should be reevaluated.

## Methods

The details of all the reagents used are listed in Supplementary Table S1.

**Fig. 5 | DP2 antagonism or DP1 agonism enhances NK cell IFN-γ production and alters the phenotype of alveolar macrophages.** Mice with CEA were inoculated with RV-1b and treated with a DP2 antagonist or a DP1 agonist daily starting from day 99. Mice were euthanized at 1, 3 and 7 dpi. **A** Number of IFN-γ expressing cells in the lungs at 7 dpi. **B** Number of IFN-γ expressing CD4⁺ T, CD8⁺ T and NK cells in the lungs at 7 dpi. **C** Number of TNF expressing cells in the lungs at 7 dpi. **D** Number of TNF expressing CD4⁺ T, alveolar macrophages (AMs) and conventional type-2 dendritic cells (cDC2s) in the lungs at 7 dpi. **E** Representative flow cytometry plots showing TNF expressing AMs in the lungs. **F** Number of total AMs in the lungs. **G** Representative flow cytometry plots showing CD86 and CD206 expression on

AMs and the total number of CD206-expressing AMs and CD86-expressing AMs. Data are presented as mean ± SEM or box-and-whisker plots showing individual data points with the boxes representing quartiles and whiskers indicating the range and are pooled data from two independent experiments showing similar results ($n = 4–8$ mice per group). Statistical significance between different time points or different groups was determined using one-way ANOVA with Dunnett's multiple comparison test. * denotes $p < 0.05$; ** denotes $p < 0.01$ and *** denotes $p < 0.001$ compared to vehicle group at 0 dpi. # denotes $p < 0.05$, ## denotes $p < 0.01$ and ### denotes $p < 0.001$ compared to RV-infected group at corresponding time point. Source data are provided as a Source Data file.

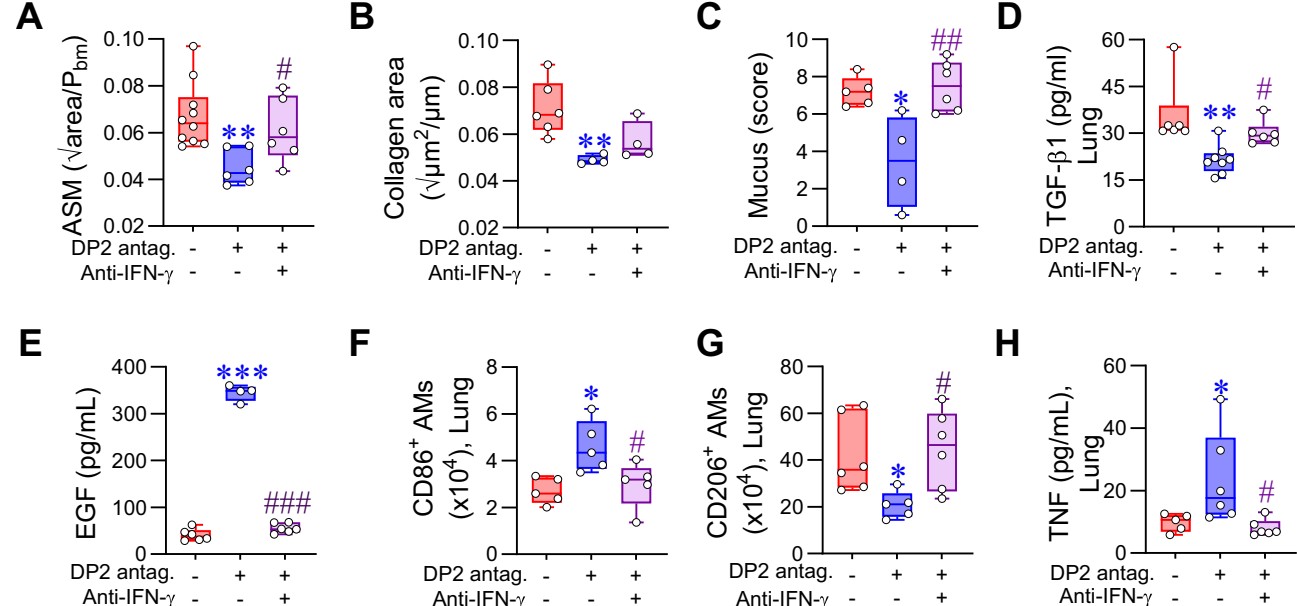

**Fig. 6 | DP2 antagonism mediates airway remodeling resolution via enhanced IFN-γ.** Mice with CEA were inoculated with RV-1b mice and treated with vehicle or DP2 antagonist +/− anti-IFN-γ. Mice were euthanized at 7 dpi. Quantification of (**A**) ASM area and (**B**) collagen deposition and (**C**) Mucus score. (**D**) Lung TGF-β1 levels in homogenates. **E** Lung EGF levels in homogenates. **F, G** Number of CD86-expressing AM and CD206-expressing AMs in the lung. **H** Lung TNF levels. Data are presented as box-and-whisker plots showing individual data points with the boxes

representing quartiles and whiskers indicating the range and are representative of two independent experiments showing similar results ($n = 5–10$ mice per group). Statistical significance between different groups was determined using one-way ANOVA with Dunnett's multiple comparison test. * denotes $p < 0.05$; ** denotes $p < 0.01$ and *** denotes $p < 0.001$ compared to vehicle group. # denotes $p < 0.05$, ## denotes $p < 0.01$ and ### denotes $p < 0.001$ compared to DP2 antagonist-treated group. Source data are provided as a Source Data file.

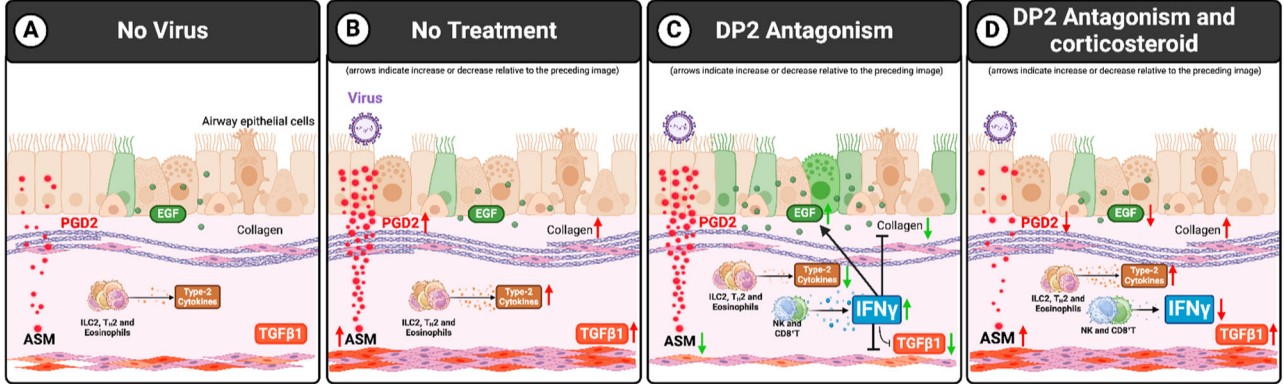

**Fig. 7 | Proposed mechanism by which DP2 antagonism resolves ASM mass.** Box **A** PGD2, EGF, and TGF-β1 are expressed at low levels in mice with stable chronic experimental asthma. Box (**B**) Upon a virus-associated exacerbation, PGD2 levels increase, elevating type-2 inflammation, TGF-β1 expression, and ASM remodeling. Box (**C**) DP2 antagonism ameliorates the aforementioned phenotypes via increased

IFN-γ production by NK and CD8⁺ T cells. Box (**D**) By suppressing the production of endogenous PGD2, corticosteroids ablate the PGD2/DP1 receptor-induced IFN-γ expression that mediates the resolution of ASM remodeling. Created in BioRender. Howard, D. (2023) BioRender.com/c86z618.

## Rhinovirus-induced exacerbation model

Neonatal BALB/c mice (male and female) were intranasally (i.n. route) inoculated with pneumonia virus of mice (PVM; 1 PFU) at postnatal day (PND) 7 and cockroach extract (CRE, 1 μg; i.n. route) at 3 days post-infection (dpi). Mice were challenged with PVM (100 PFU) at 42 dpi and CRE (1 μg) at 45, 52, 59 and 66 dpi to induce CEA[23]. Four weeks later, mice were inoculated with vehicle or RV-1b ($5 \times 10^6$ median tissue culture infective dose [TCID50]) to induce an acute exacerbation (Fig. 1A)[24,25,49]. Where indicated, separate groups of mice were treated with diluent; a DP1-specific agonist, BW245c (1 mg/kg; i.n.)[19,27,50,51]; a DP1-specific antagonist, MK-0524 (5 mg/kg; i.n.)[19,52]; a DP2-specific antagonist, timapiprant, also known as OC000459 (10 mg/kg; oral gavage)[53]; or a hematopoietic prostaglandin D synthase (h-PGDS) inhibitor, PK007, generated in the laboratory of Dr. Mark Smythe (10 mg/kg; oral gavage)[54] from postnatal day 99 until euthanasia. In some experiments, mice were treated with soluble IL-13Rα2 (i.n., 200 μg/dose; Pfizer, NY, USA) on alternate days starting from postnatal day 100 or fluticasone (i.n.; 1 mg/kg; Sigma-Aldrich, MO, USA)[55] daily from postnatal day 99 onwards, and the mice euthanized at 1, 3, and 7 dpi. Where indicated, anti-IFN-γ (8 mg/kg; i.p.)[56] antibody was administered on alternate days starting at postnatal day 100. All the mice were housed on a 12-hour light/12-hour dark cycle at 20-24$^0$C temperature with 55% humidity. All studies were approved by the Animal Ethics Committee of QIMR Berghofer Medical Research Institute. Mice were euthanized at the indicated time points, and different tissue samples were harvested and processed for immunological assays as outlined in the supplementary section.

## Lung tissue harvesting and processing

Following euthanasia, bronchoalveolar lavage (BAL) was performed. The cell-free supernatant was stored at −80 °C prior to analysis. The left lobe was then immediately processed for flow cytometry analysis. The superior right lobe was placed in neutral buffered formalin for fixation and processed for histological analysis. The right inferior lobe was snap frozen in protein assay buffer containing sodium chloride (150 mM), IGEPAL® CA-630 (1%), sodium deoxycholate (0.5%), sodium dodecyl sulfate (0.1%) and Tris buffer (50 mM) and stored −80 °C. This lobe was later homogenized with a tissue-tearor (Biospec, OK, USA), centrifuged at 13,000 rpm for 10 min, and the supernatant harvested and stored at −80 °C for ELISA.

## Flow cytometry

Single-cell suspension was prepared from lung tissue as described previously[57]. Briefly, lung lobes were mechanically dissociated by pressing through 70 μm cell strainers and the single cell suspension was prepared in FASC buffer (phosphate-buffered saline (PBS) supplemented with 2% fetal calf serum, FCS). Red blood cells were lysed with Gey's lysis buffer, and the cells were re-suspended and washed in FASC buffer. Cells were incubated with 2.4G2 antibody for 30 min at 4 °C to prevent non-specific binding, then stained with the surface antibody cocktail for 30 min at 4 °C. The list of antibodies used for flow cytometry is summarized in Supplementary Table S1. To detect intracellular transcription factors, the cells were first stained with surface antibodies then permeabilized and fixed using FoxP3/transcription factor fixation/permeabilization kit (eBiosciences; CA, USA). After washing with permeabilization buffer, the cells were stained with fluorochrome-conjugated antibodies against ROR-γt, GATA3 or Tbet. For intracellular cytokine staining, cells were stimulated with phorbol 12-myristate 13-acetate (PMA; 50 ng/ml) and ionomycin (1 μg/ml) in the presence of brefeldin A (20 μg/ml) at 37 °C for 3 h, then washed with FACS buffer and stained with surface antibodies as above. Cells were permeabilized and fixed using FoxP3/transcription factor fixation/permeabilization kit and stained with appropriate antibodies. The samples were acquired on a BD Fortessa IV flow cytometer (BD Biosciences, USA) using the FACSDiva software (version 8, BD Biosciences, USA). Data was analyzed using FlowJo software (Version 10.6; TeeStar, USA) and the gating strategy has been depicted in Supplementary Fig. S4.

## Histology

Paraffin-embedded lung tissues were trimmed until the large and small airways were visible and then five consecutive sections (5 μm thick) were cut for immunohistochemistry, which was performed as previously described[58,59]. Briefly, lung sections were deparaffinized and rehydrated, then immersed in citrate buffer and heated in a pressure cooker for antigen retrieval. Sections were permeabilized using 0.6% Tween-20 and then incubated with 10% goat serum in PBS for thirty minutes at room temperature. Subsequently, sections were incubated with the following primary antibodies (details are in Supplementary Table S1): mouse anti-Muc5ac (1:400), mouse anti-α-Smooth muscle actin (α-SMA; 1:800), rabbit anti-TGF-β1 (1:400) or rat anti-EGF (1:10). The following day, sections were washed and incubated with the appropriate alkaline-phosphatase secondary antibody. The color was developed with fast-red reagent. Sections were counter-stained with hematoxylin and mounted with glycerol gel. Tissue was stained with picrosirius red (PSR) to visualize collagen expression. All the stained slides were scanned using Aperio AT Turbo (Leica Biosystems, Wetzlar, Germany). Muc5ac staining was scored on a scale of 1–5 for the percentage of airway epithelial cells expressing Muc5ac and 1–5 for the percentage of the airway with Muc5ac plugging with a maximum score of 10[60]. ASM, collagen, EGF and TGF-β1 expression was assessed around the airways with a circumference <700 μm. Five airways were assessed per mouse, biasing towards the airways with greatest immunoreactivity in all cases[60]. The specificity of EGF immunoreactivity was confirmed by preincubation of the antibody with exogenous EGF. The sum of ASM or collagen area around the small airways was quantified using Aperio ImageScope software (Leica Biosystems, Wetzlar, Germany) and the square root of the ASM or collagen area per micrometer of basement membrane circumference was calculated[23,25,58,60,61]. EGF expression was calculated as a percentage of immunoreactive-positive AECs. TGF-β1 expression was expressed as the percentage of the airway perimeter with adjacent TGF-β1 expressing cells[58]. The mean value of five airways for each mouse was calculated and plotted for each parameter. All the analysis was performed by trained staff in a blinded manner without knowledge of the experimental conditions.

## ELISA assay

The expression of cytokines, chemokines, growth factors, and PGD2 was measured using commercially available kits as per the manufacturer's instructions (Supplementary Table S1).

## Statistical analysis

All the experiments were performed at least two times. Where data are presented as box-and-whisker plots, the boxes represent quartiles and whiskers indicate the range. The software GraphPad Prism (version 8, USA) was used for statistical analysis. Group differences were analyzed by the Mann−Whitney U test or one-way ANOVA with Dunnett's multiple comparison tests as appropriate. Significance was set at a $P$ value less than 0.05 for all tests.

## Reporting summary

Further information on research design is available in the Nature Portfolio Reporting Summary linked to this article.

# Data availability

All data are available in the article and its Supplementary files or from the corresponding author upon request. Source data are provided with this paper.

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

## Acknowledgements

This work was supported by National Health and Medical Research Council (NHMRC, Australia) grants awarded to S.P. S.R. was funded by the doctorate program MOLIN (FWF, W1241) and an EMBO Short-Term Fellowship (8109). We thank all the staff from the QIMR Berghofer animal facility, flow cytometry laboratory and histology facility for their assistance, and Christina Kulis (Institute for Molecular Bioscience, University of Queensland) for the supply of the h-PGDS inhibitor, PK007.

## Author contributions

M.A.U. and S.P. conceived the project, designed the experiments, analyzed and interpreted the data and wrote the manuscript. M.A.U., S.R., J.L. and B.F.C. conducted experiments, analyzed and interpreted the data. P.N., T.A., D.R.H., M.M.R., M.A.A.S., R.B.R., N.C. and M.L. performed experiments and analyzed the data. M.L.S. provided critical reagents and intellectual input. All the authors reviewed and approved the manuscript.

## Competing interests

S.P. has performed contract work with Novartis and received speaker and consultancy fees from Novartis. M.L.S. is an employee of Infensa Bioscience. The rest of the authors have no relevant conflicts of interest.
