## [Transparent Peer Review file · Nature Communications]

Dual therapy with corticosteroid ablates the beneficial effect of DP2 antagonism in chronic experimental asthma

Corresponding Author: Professor Simon Phipps

Version 0:

Reviewer comments:

Reviewer #1

(Remarks to the Author)

In this manuscript Ullah and colleagues investigate the role of PGD2 in a mouse model of virus-induced asthma exacerbations. Extensive pre-clinical research to date indicates that PGD2 contributes to asthma pathophysiology, primarily via activation of DP2 (CRTh2). Several DP2 antagonists have been developed for use in human subjects, but the recent clinical experience with these compounds in phase III studies was disappointing. Here the authors test one possibility explaining the recent experience with DP2 antagonists in clinical trials, namely that their efficacy might be attenuated by the concomitant use of glucocorticoids (which are commonly used by severe asthmatics). Using a mouse model of virus induced asthma exacerbations, the authors report that the steroid fluticasone attenuated the beneficial effects of a DP2 antagonist on some parameters of airway remodelling including airway smooth muscle (ASM) mass. They identified a potential mechanism for the beneficial effects of DP2 antagonism on ASM remodelling in virus infected mice involving enhanced IFN γ production probably from NK cells, which was also attenuated by concomitant use of fluticasone.

This manuscript has several strengths and a some weaknesses. Strengths include the clinical / translational relevance of the model, a well-written and well-referenced manuscript, and the comprehensive use of multiple approaches including pathway agonists / antagonists, flow cytometry, and a neutralizing anti-IFN γ antibody. The use of a mouse model of chronic asthma with virally-mediated exacerbation is innovative. Taken together, the data shown generally support the proposed model. Although there is precedence for the idea that DP2 antagonism reduces ASM mass in human asthmatics (Saunders et al, Ref 11), this manuscript suggests a potential mechanism for the observed effects. Additionally Figure 1 shows that ASM remodeling can revert very quickly after initiating treatment with a DP2 antagonist (3-7 days after viral infection): if replicated by others, this observation is potentially ground breaking.

There are a few weaknesses however, some conceptual and some methodological.

This reviewer is having a hard time understanding how DP2 antagonism only works to reduce ASM mass in the presence of RV infection (Figure 1) and is completely ineffective in the absence of RV infection (Figure S1). The authors speculate that this reflects "RV-induced PGD2 release", but their data show that: (i) there is constitutive PGD2 in the lung (Figure 1B), and (ii) ASM remodelling occurs before RV infection. More discussion of the amount of PGD2 produced before and after RV infection, and how this relates to predicted PGD2 receptor engagement taking into account receptor expression and affinity, would seem warranted. Please explain this further in the manuscript and include a panel of "No virus" in the schematic model.

Another conceptual problem is that the human clinical study (Saunders et al, Ref 11) showed that the DP2 antagonist fevipiprant reduced ASM mass in human subjects with severe asthma, all of whom were using inhaled steroids. This seems to challenge a central premise of this paper, namely that DP2 antagonists will only "work" in the absence of concomitant steroids. Please explain.

One strength of this study is the use of a mouse model of virus-induced asthma exacerbation. Does RV-1b replicate in the model system, and if so any reason to suspect attenuated RV1-b replication in the presence of DP2 antagonists, similar to their recent work with RSV?

A methodological concern is that the key readout of ASM mass (alpha-SMA staining normalized to epithelial basement

membrane) is semi-quantitative and subject to sampling error. No other readouts of ASM hyperplasia/hypertrophy or airway physiology were provided. Please explain how many and what size airways were chosen for analysis per mouse in this regard, and what approaches were taken to standardize histological sectioning and ensure objective analysis. Were these analyses carried out blinded to experimental condition: if not, why not?

Please explain the rationale for doses of the different agonists, antagonists and fluticasone used in the model, how they relate to those used in human clinical studies / practice, and how the authors monitored or controlled for off-target effects.

From the data presented in the Figures and Figure legends, it is not possible to discern the variability or repeatability in the responses shown. The data shown are either pooled (Figures 1, 2, 5) or representative (Figures 3 and 4) with variable mice numbers, and graphs depict mean \pm SEM or box and whisker plots. It would help the reader understand number of replicates per condition, as well as variability in responses, to depict individual data points in the graphs.

Please confirm spelling of two authors first names ("Md Ashik" and "Md Al Amin")?

Reviewer #2

(Remarks to the Author)

In this manuscript, Ullah et al studied the role of Prostaglandin D2 (PGD2) signaling via the DP1 and DP2 receptors in asthma, particularly focusing on the impact of DP2 antagonism on asthma exacerbations and ASM remodeling. Interestingly, they found that treatment with a DP2 antagonist or DP1 agonist alleviated asthma-related phenotypes and decreased ASM area. In contrast, treatment with dual DP1-DP2 antagonism or the combination of corticosteroid and DP2 antagonist showed reduced endogenous PGD2 levels, preventing ASM resolution. They provide an interesting rationale that the timing and combination of treatments can impact the effectiveness of DP2 antagonists in managing asthma exacerbations and ASM remodeling. Overall, the passage presents intriguing research findings with potential implications for asthma treatment, but it would benefit from improved clarity, novelty, additional evidence, and a broader field of asthma research.

1: While this study focuses on airway remodeling in asthma, similar findings were reported by the same research group in their study on RSV bronchiolitis entitled "PGD2/DP2 receptor activation promotes severe viral bronchiolitis by suppressing IFN- γ production" (Sci Transl Med 2018; 10 (440)). Especially, they have suggested that DP2 antagonists or DP1 agonists are a useful treatment against viral bronchiolitis and a primary preventive against asthma development. Mechanistically, they explored immune response modulation, particularly the role of type-1 immunity (IFN- γ and IFN- λ). Thus, the concern about novelty is raised in light of the journal's reputation, Nature Communications.

2. The study effectively showed that combining dual DP1-DP2 antagonism or corticosteroid with a DP2 antagonist hindered the effectiveness of DP2 antagonism in managing asthma exacerbations and ASM remodeling. However, there is a need for further investigation as the study lacks direct evidence to substantiate their conclusions. Specifically, the mechanisms by which these dual treatments suppress endogenous PGD2 and IFN- γ production, leading to the prevention of ASM resolution, require additional clarification and supporting evidence.

3. It is indeed clear that PGD2/DP2 plays a role in immune response modulation, but the mechanisms behind how PGD2/DP1 signaling regulates Th1 immunity and airway remodeling remain less understood. DP1's distribution in various tissues, especially in blood vessels and airway smooth muscles, contributes to vasodilation and bronchodilation. Therefore, it's important to consider multiple layers of cross-talk in response to PGD2, involving interactions between immune cells and structural cells (such as airway smooth muscles) and between DP1 and DP2 receptors. Further research is needed to elucidate the complex interactions and signaling pathways involved in airway remodeling as a secondary event to inflammation.

4. Numerous markers have been implicated in airway remodeling, including PDGF, TGF β , EGF, MMPs, and cytokines (e.g., IL-33). It's important to consider why only TGF β was monitored (see Figure 2).

5. It should be expanded or explained for the methods used to quantify ASM or collagen areas (e.g., varea/Pbm).

6. For consistency, it would be of interest to add a group in experiments presented in Figure 5, DP2/DP1 antagonist or DP2/Fut group to see whether these enhanced cell types can be reversed as a major cellular source of IFN- γ . Also in Figure 5, both the number of IFN- γ + cells in Figure 5A and B or TNF- α + cells in Figure 5C and D should be expressed consistently (x10⁴).

Reviewer #3

(Remarks to the Author)

This manuscript examines the consequences of dual therapy with a steroid and DP2 antagonism in allergic airways disease with exacerbation. However, there are a few points regarding experimental design and discussion/interpretation that should be clarified:

1. The authors postulate that the effects of DP2 antagonism are mediated through DP1 activation and PGD2 production. Why is it that a DP1 antagonist increased PGD2 release into the lung (Figure 1B) similarly to the DP2 antagonist? It doesn't make sense that antagonism of DP1 would then lead to PGD2 release, it should actually inhibit it instead if we follow the aforementioned hypothesis. The data presented in Figure 1 doesn't make sense with the data in Figure 3 because there should have been the same decreases in ASM area, collagen, and mucus production with the DP1 antagonist as seen in Figure 1, but that wasn't the case. The DP1 antagonist actually showed no change compared to vehicle in most of the parameters, or a modest increase in some of them instead of a decrease.

2. The representative lung pathology images show very little morphologic changes between naïve and asthma in both the absence and presence of RV infection, so it's hard to reconcile the collagen area changes since they don't look grossly abnormal in comparison to the naïve mice. There doesn't look to be much airway wall thickening, which is a great indicator of airway remodeling.

Minor points for consideration:

3. Check Figure 6C, as there is text over the top of the x axis label that obscures the label.

4. Despite the findings of this study concerning the effects of corticosteroids almost negating the effect of the DP2 antagonism, I feel that it will be difficult to find an asthmatic population that wouldn't be on corticosteroids of some sort to actually test whether the DP2 antagonism would be more effective in the absence of steroids.

5. It would have been good to also have lung function parameters on the mice to determine whether the interventions used had an effect in the absence of steroids, which would've strengthened the data.

Version 1:

Reviewer comments:

Reviewer #1

(Remarks to the Author)

The authors have adequately addressed the prior critique in my opinion.

Reviewer #2

(Remarks to the Author)

My concerns have been well-addressed. The new findings relating to epithelial EGF expression certainly strengthens the manuscript.

Reviewer #3

(Remarks to the Author)

The clarification of data and methods, as well as inclusion of additional data makes this a strong manuscript.

REVIEWER COMMENTS

Reviewer #1 (Remarks to the Author):

In this manuscript Ullah and colleagues investigate the role of PGD2 in a mouse model of virus-induced asthma exacerbations. Extensive pre-clinical research to date indicates that PGD2 contributes to asthma pathophysiology, primarily via activation of DP2 (CRTh2). Several DP2 antagonists have been developed for use in human subjects, but the recent clinical experience with these compounds in phase III studies was disappointing. Here the authors test one possibility explaining the recent experience with DP2 antagonists in clinical trials, namely that their efficacy might be attenuated by the concomitant use of glucocorticoids (which are commonly used by severe asthmatics). Using a mouse model of virus induced asthma exacerbations, the authors report that the steroid fluticasone attenuated the beneficial effects of a DP2 antagonist on some parameters of airway remodelling including airway smooth muscle (ASM) mass. They identified a potential mechanism for the beneficial effects of DP2 antagonism on ASM remodelling in virus infected mice involving enhanced IFN γ production probably from NK cells, which was also attenuated by concomitant use of fluticasone.

This manuscript has several strengths and a some weaknesses. Strengths include the clinical / translational relevance of the model, a well-written and well-referenced manuscript, and the comprehensive use of multiple approaches including pathway agonists / antagonists, flow cytometry, and a neutralizing anti-IFN γ antibody. The use of a mouse model of chronic asthma with virally-mediated exacerbation is innovative. Taken together, the data shown generally support the proposed model. Although there is precedence for the idea that DP2 antagonism reduces ASM mass in human asthmatics (Saunders et al, Ref 11), this manuscript suggests a potential mechanism for the observed effects. Additionally Figure 1 shows that ASM remodeling can revert very quickly after initiating treatment with a DP2 antagonist (3-7 days after viral infection): if replicated by others, this observation is potentially ground breaking.

There are a few weaknesses however, some conceptual and some methodological.

This reviewer is having a hard time understanding how DP2 antagonism only works to reduce ASM mass in the presence of RV infection (Figure 1) and is completely ineffective in the absence of RV infection (Figure S1). The authors speculate that this reflects "RV-induced PGD2 release", but their data show that: (i) there is constitutive PGD2 in the lung (Figure 1B), and (ii) ASM remodelling occurs before RV infection. More discussion of the amount of PGD2 produced before and after RV infection, and how this relates to predicted PGD2 receptor engagement taking into account receptor expression and affinity, would seem warranted. Please explain this further in the manuscript and include a panel of "No virus" in the schematic model.

Reply #1:

We thank the reviewer for their positive and insightful comments. As the reviewer correctly points out, DP2 antagonism is insufficient to resolve ASM remodelling even though there is a PGD2 signal at steady-state. As PGD2 affinity for DP1 and DP2 is almost identical¹, this suggests that the low levels of basal PGD2 levels are insufficient

and/or that a secondary signal is required to mediate the remodelling, at least in the time frame employed in the model. As highlighted in Fig 6, the effect of DP2 antagonism is lost following neutralisation of IFN- γ , which is produced by NK cells, T cells, and other immune cells that infiltrate the lung mucosa in response to RV infection. Thus, RV-induced **cellular inflammation** is a necessary component for PGD2/DP1-mediated resolution of airways remodelling.

Based on the reviewer's comment, we have edited the manuscript in two places to increase its clarity. In the results section, we have altered the text (bold) to state:

"To test this, we hypothesized that the beneficial effects of DP2 antagonism (i) would not occur in the absence of RV-induced **inflammation**" (previously RV-induced PGD2 release').

And in the Discussion, we have added the following text:

"Our findings revealed that DP2 antagonism in the absence of a RV-induced exacerbation has no benefit and that dual DP2 antagonism/steroid therapy is ineffective, **suggesting that immune cells actively contribute and are necessary for the beneficial effects of DP2 antagonism**".

With regards to the graphical abstract, this has been revised as per the reviewer's instruction, and we have also included EGF in the GA in light of our new findings.

Another conceptual problem is that the human clinical study (Saunders et al, Ref 11) showed that the DP2 antagonist fevipiprant reduced ASM mass in human subjects with severe asthma, all of whom were using inhaled steroids. This seems to challenge a central premise of this paper, namely that DP2 antagonists will only "work" in the absence of concomitant steroids. Please explain.

Reply#2:

The reviewer makes an important point and we touched on this ourselves in the discussion. The effectiveness of the DP2 antagonists cannot be simply explained by steroid use since all of the patients should have been taking steroids, although it is worth noting that the dose and route (inhaled versus oral) would have varied, and that compliance is a major issue in asthma. Therefore, the effectiveness of the DP2 antagonist (in each individual) most likely relates to other factors, such as the underlying endotype, the inflammatory microenvironment, and responsiveness to steroid treatment, which is known to be highly variable in asthma, e.g. steroid insensitive, steroid resistant or refractory asthma. Indeed, on inspection of the data from the Phase II clinical trial where the response for each individual is presented, it is clear that there is marked heterogeneity in responsiveness to treatment.^{2, 3} Of the patients who received active treatment, only 9 of 14 showed a decrease in ASM area. In 2 patients, ASM did not change, in 2 patients ASM increased, and it is not possible to discern the outcome in the remaining patient. In the placebo group, 3 of 13 patients showed a decrease in ASM mass (i.e. 64.3% active vs 23% placebo). The patients in this trial had moderate to severe asthma, and an elevated sputum eosinophilia, but otherwise there are limited details in the study design section of the Materials and Methods describing medication use, or steroid sensitivity. The effect of DP2

antagonism on endophenotypical pathways was not explored, although the authors did employ computational modelling, which implicated a role for decreased numbers of eosinophils (which we also observed in our model) and importantly they also concluded 'the existence of additional mechanisms' in mediating the reduction in ASM mass after treatment with fevipiprant. With the advent of single cell technology and spatial transcriptomics, it would be fascinating to re-evaluate the lung biopsies to discern novel molecular pathways that underpin drug effectiveness and to help explain the heterogeneity of drug responsiveness.

As noted in the discussion, our findings suggest that patients with steroid insensitive disease would be more likely to benefit from treatment with a DP2 antagonist since this would have allowed for greater levels of endogenous PGD2 to be produced upon a viral infection or other trigger of PGD2 (e.g. allergen). In this regard, it is pertinent to note that in the Phase II study performed by Saunders et al, the investigators state that some of the patients in the trial had refractory eosinophilic asthma, which can be associated with more severe inflammation, greater exacerbations, steroid resistance, and aspirin sensitivity⁴ the latter linked to elevated PGD2 levels^{5,6} This is noteworthy since each of these phenotypes would favour drug efficacy via increased cellular inflammation and/or elevated PGD2 levels (and hence downstream DP1 activation) necessary for DP2 antagonist-mediated protection.

In light of the reviewer's comment, our comments above, and the new findings relating to epithelial EGF expression, we have now substantially revised the discussion, which we feel is now much improved.

One strength of this study is the use of a mouse model of virus-induced asthma exacerbation. Does RV-1b replicate in the model system, and if so any reason to suspect attenuated RV1-b replication in the presence of DP2 antagonists, similar to their recent work with RSV?

Reply#3: Human RV replicates extremely poorly in mice and viral load does not drive pathogenicity in mice. We and others have shown that RV is detectable in the lungs at 1 dpi but not 3 dpi,^{7,8} and so it is highly unlikely that the beneficial effect of DP2 antagonism relates to accelerated clearance. Moreover, in the setting of chronic established asthma, we did not observe a beneficial effect of DP2 antagonism on IFN-lambda production (data not shown).

A methodological concern is that the key readout of ASM mass (alpha-SMA staining normalized to epithelial basement membrane) is semi-quantitative and subject to sampling error. No other readouts of ASM hyperplasia/hypertrophy or airway physiology were provided. Please explain how many and what size airways were chosen for analysis per mouse in this regard, and what approaches were taken to standardize histological sectioning and ensure objective analysis. Were these analyses carried out blinded to experimental condition: if not, why not?

Reply#4: We apologise for not including sufficient information in this regard; we instead referenced previous articles by our group to keep under the word limit. We

have previously identified that ASM area in moderate-sized and small airways is affected in response to viral or allergen challenges, and that this response is highly reproducible.^{8, 9, 10, 11, 12, 13} Consequently, we assess ASM area in airways with a circumference of less than 700 μm and select the 5 airways (per animal) with the greatest amount of ASM mass to generate a mean value for each mouse. The superior right lobe of each mouse is embedded in paraffin with the bronchus entering side facing up for sectioning. The tissue is trimmed off until the large airway is visible and then 6 consecutive sections are collected for the various histological analyses. The analysis is then performed by experienced investigators in a blinded manner without knowledge of the experimental conditions, and consistent with our previous papers.^{8, 9, 10, 11, 12, 13} We have now updated the Histology section in the Methods to improve its clarity.

Please explain the rationale for doses of the different agonists, antagonists and fluticasone used in the model, how they relate to those used in human clinical studies / practice, and how the authors monitored or controlled for off-target effects.

Reply#5: The in-vivo doses of different reagents were selected based on published reports. We have now cited the relevant references in the methods section. In clinical trials,^{2, 3, 14} a daily 450 mg tablet of DP2 antagonist fevipiprant was given to patients. In the LUSTER studies,¹⁴ the mean age was 50 years and a healthy 50 years old person will weigh around 75 kg. Therefore, the patients were dosed at approximately 6 mg/kg body weight. Based on this information and previously published pre-clinical studies with OC000459¹⁵, we treated at 10 mg/kg daily. We acknowledge that drugs often have off-target effects, however extensive pharmacology has been undertaken on the DP2 antagonists and these are highly selective for DP2 at the dose used.¹⁵ It is worth noting that in our previous paper published at Sci Transl Med that we used a different DP2 antagonist (AM156)¹³ and observed a similar phenotype, providing greater confidence that the drugs are acting through the postulated mechanism.

From the data presented in the Figures and Figure legends, it is not possible to discern the variability or repeatability in the responses shown. The data shown are either pooled (Figures 1, 2, 5) or representative (Figures 3 and 4) with variable mice numbers, and graphs depict mean \pm SEM or box and whisker plots. It would help the reader understand number of replicates per condition, as well as variability in responses, to depict individual data points in the graphs.

Reply#6: For the box and whisker plots, we have now added the individual data points as per the reviewer's request. However, the time course studies are best viewed as XY plots (mean \pm SEM) as presenting these data separately for each time point decreases the ease of understanding and interpretation of the paper. Therefore, we have left the graphs with time course data as they are.

Please confirm spelling of two authors first names ("Md Ashik" and "Md Al Amin")?

Reply#7: Thank you for checking. Both of these first names are spelt correctly.

Reviewer #2 (Remarks to the Author):

In this manuscript, Ullah et al studied the role of Prostaglandin D2 (PGD2) signaling via the DP1 and DP2 receptors in asthma, particularly focusing on the impact of DP2 antagonism on asthma exacerbations and ASM remodeling. Interestingly, they found that treatment with a DP2 antagonist or DP1 agonist alleviated asthma-related phenotypes and decreased ASM area. In contrast, treatment with dual DP1-DP2 antagonism or the combination of corticosteroid and DP2 antagonist showed reduced endogenous PGD2 levels, preventing ASM resolution. They provide an interesting rationale that the timing and combination of treatments can impact the effectiveness of DP2 antagonists in managing asthma exacerbations and ASM remodeling. Overall, the passage presents intriguing research findings with potential implications for asthma treatment, but it would benefit from improved clarity, novelty, additional evidence, and a broader field of asthma research.

1: While this study focuses on airway remodeling in asthma, similar findings were reported by the same research group in their study on RSV bronchiolitis entitled "PGD2/DP2 receptor activation promotes severe viral bronchiolitis by suppressing IFN- γ production" (Sci Transl Med 2018; 10 (440)]. Especially, they have suggested that DP2 antagonists or DP1 agonists are a useful treatment against viral bronchiolitis and a primary preventive against asthma development. Mechanistically, they explored immune response modulation, particularly the role of type-1 immunity (IFN- γ and IFN- λ). Thus, the concern about novelty is raised in light of the journal's reputation, Nature Communications.

Reply#8: We respectfully disagree. Our previous study focused on how PGD2/DP2 signalling suppresses IFN- λ and impairs antiviral immunity in the setting of a neonatal acute lower respiratory infection, whereas, the current study investigates how DP2 antagonist modulates chronic airway remodelling and inflammation in established disease. Additionally, we provide novel insights into the mechanism by which DP2 antagonism reverses airway remodelling, and the data provides a potential explanation for the failure of DP2 antagonists in the Phase II/III trials – namely the use of corticosteroids. These findings have significant clinical implications as pointed out by reviewer 1. Of note, fevipiprant has been out-licensed from Novartis, and DP2 antagonists are being explored in the context of tumour immunity (in particular where a type-2 environment is immunosuppressive) and COPD, so we are confident that our manuscript is novel and will be of broad interest to the scientific community.

2. The study effectively showed that combining dual DP1-DP2 antagonism or corticosteroid with a DP2 antagonist hindered the effectiveness of DP2 antagonism in managing asthma exacerbations and ASM remodeling. However, there is a need for further investigation as the study lacks direct evidence to substantiate their conclusions. Specifically, the mechanisms by which these dual treatments suppress endogenous PGD2 and IFN- γ production, leading to the prevention of ASM resolution, require additional clarification and supporting evidence.

Reply#9: We observed that IFN- γ levels were lower in mice with CEA compared to naïve controls. RV inoculation led to a small increase in IFN- γ levels in CEA mice (Fig. 4E), and this IFN-g response was greater at 1, 3 and 7 dpi (reaching significance at 3 and 7 dpi) when the mice were treated with the DP2 antagonist or DP1 agonist (Fig. 4E). We demonstrated that fluticasone decreased PGD2 levels (Fig. 3J) consistent with the literature describing the inhibitory effect of steroids on phospholipase A2 and ability to downregulate COX2 and PGD2 expression,^{16, 17, 18} and provided links to the literature in the results section. We have now edited this section to note the inhibitory effects of steroids on phospholipase A2 as follows:

“The requirement for endogenous PGD2 suggested that the effectiveness of DP2 antagonism would be decreased by co-treatment with a corticosteroid, as corticosteroids **inhibit phospholipase A2, and lower the expression of cyclooxygenases, thereby decreasing prostanoid production**”.

Our conclusions are substantiated by our data: DP1 antagonism naturally blocks the effect of PGD2 at the DP1 receptor, and accordingly, IFN- γ was no longer increased. As noted, steroids act upstream of this (PGD2/DP1 signaling) by decreasing the production of PGD2, and thus, indirectly ablate the PGD2/DP1-induced increase in IFN- γ production. It is likely that the corticosteroid treatment additionally affected the NK/T cell response since steroids elicit broad spectrum anti-inflammatory effects, as is widely known. We demonstrated that fluticasone decreased both PGD2 and IFN-g production, and that both PGD2 and IFN-g were critical for ASM resolution. In our view, evaluating the specific mechanism (of which there could be many) by which steroids decreased PGD2 and IFN-g would not improve the impact of the manuscript since this would not be impactful and is not the main thrust of the paper; rather, the paper's novelty stems from the observation that dual steroid/DP2 antagonism ablates the beneficial effects of DP2 antagonism, which are mediated by PGD2/DP1 signalling, and downstream, enhanced IFN-g production (and now, based on our additional findings, potentially a role for EGF).

3. It is indeed clear that PGD2/DP2 plays a role in immune response modulation, but the mechanisms behind how PGD2/DP1 signaling regulates Th1 immunity and airway remodeling remain less understood. DP1's distribution in various tissues, especially in blood vessels and airway smooth muscles, contributes to vasodilation and bronchodilation. Therefore, it's important to consider multiple layers of cross-talk in response to PGD2, involving interactions between immune cells and structural cells (such as airway smooth muscles) and between DP1 and DP2 receptors. Further research is needed to elucidate the complex interactions and signaling pathways involved in airway remodeling as a secondary event to inflammation.

Reply#10: We acknowledge that both haematopoietic and non-haematopoietic cells respond to PGD2, and this is noted in the manuscript where relevant. Additional layers of complexity in disentangling the key cellular players and pathways in what is a highly dynamic tissue microenvironment include the labile nature of PGD2, the generally opposing nature of the receptors (DP1 and DP2) in terms of biological outcomes, and the contribution of PGD2 metabolites, which are also able to activate DP1 and DP2. We cannot study all of these interactions, even in a preclinical model. Ultimately, we primarily focused on the immune system since the neutralisation of IFN-g ablated the

protective effect of DP2 antagonism on airway remodelling. We have previously shown that DP1 agonism (in vitro) induces CD4 T cells to produce TNF (ref 27 in the manuscript), i.e. PGD₂/DP1 signalling occurs directly on the affected lymphocytes. Both AECs and ASM cells express the receptor for IFN- γ , and therefore future studies will need to interrogate whether IFN- γ mediates the decrease in ASM mass indirectly (e.g. by increasing EGF) or directly (e.g. by decreasing TGF- β), or both, which we note in the discussion.

4. Numerous markers have been implicated in airway remodeling, including PDGF, TGF β , EGF, MMPs, and cytokines (e.g., IL-33). It's important to consider why only TGF β was monitored (see Figure 2).

Reply#11: Our original study was hypothesis-based. TGF- β is the most potent ASM mitogen and widely linked to increased ASM hypertrophy and hyperplasia in the context of asthma, and therefore formed the focus of our study. However, in an attempt to address the reviewer's question, we measured several other cytokines that have been linked to the pathogenesis of asthma and airway remodelling, including IL-33, IL-25, IL-1 β , IL-6, amphiregulin and epidermal growth factor (EGF). Of these, the findings with EGF were particularly interesting since its expression in lung homogenates was elevated by both DP2 antagonism and DP1 agonism, and inversely associated with ASM area. Intriguingly, the pattern of EGF immunoreactivity in the epithelium mirrored that observed in the lung homogenates, and significantly, the DP2 antagonist induced increase in EGF expression was ablated by (i) dual DP1/DP2 antagonism, (ii) dual fluticasone/DP2 antagonism, and (iii) dual DP2 antagonism/anti-IFN- γ , implicating EGF as a key factor that promotes ASM resolution. We thank the reviewer for their comment as these data have revealed a novel axis and the revised manuscript, with these new data, is much improved.

5. It should be expanded or explained for the methods used to quantify ASM or collagen areas (e.g., varea/Pbm).

Reply#12: We apologise that the methods were not sufficiently detailed in the previous version of the manuscript. We cited our previous papers to keep the word count to a minimum. Please see our detailed reply (#5) to reviewer 1.

6. For consistency, it would be of interest to add a group in experiments presented in Figure 5, DP2/DP1 antagonist or DP2/Fut group to see whether these enhanced cell types can be reversed as a major cellular source of IFN- γ . Also in Figure 5, both the number of IFN- γ + cells in Figure 5A and B or TNF- α + cells in Figure 5C and D should be expressed consistently ($\times 10^4$).

Reply#13: Regards the second part of the reviewer's comment, we have now standardised the Y axis units. Regards the first part, regrettably this experiment would take a minimum of 5 months to perform given the requirement for animal ethics approvals, import of breeding-age mice (we do not hold a colony of BALB/c mice), initiation of timed matings/gestation, and then the extensive length of the model (>2

months) and downstream analysis. As we do not believe this experiment to be critical to the overall conclusions of the paper, and to keep the rebuttal timely, we elected not to perform this experiment. Instead, we prioritised the reviewer's comment about other remodelling factors (reviewer 2, point 4, reply 11). Our findings initially yielded negative findings, but then the EGF phenotype came to light. As we then elected to identify the cellular source by IHC in all of the models and interventions, this took a substantial amount of time, and delayed our re-submission, but we believe the new findings are of greater importance.

Reviewer #3 (Remarks to the Author):

This manuscript examines the consequences of dual therapy with a steroid and DP2 antagonism in allergic airways disease with exacerbation. However, there are a few points regarding experimental design and discussion/interpretation that should be clarified:

1. The authors postulate that the effects of DP2 antagonism are mediated through DP1 activation and PGD2 production. Why is it that a DP1 antagonist increased PGD2 release into the lung (Figure 1B) similarly to the DP2 antagonist? It doesn't make sense that antagonism of DP1 would then lead to PGD2 release, it should actually inhibit it instead if we follow the aforementioned hypothesis. The data presented in Figure 1 doesn't make sense with the data in Figure 3 because there should have been the same decreases in ASM area, collagen, and mucus production with the DP1 antagonist as seen in Figure 1, but that wasn't the case. The DP1 antagonist actually showed no change compared to vehicle in most of the parameters, or a modest increase in some of them instead of a decrease.

Reply#14: Please note that we employed a DP1 agonist and not a DP1 antagonist. As per the editor's comment, we believe the reviewer's concerns have been clarified.

2. The representative lung pathology images show very little morphologic changes between naïve and asthma in both the absence and presence of RV infection, so it's hard to reconcile the collagen area changes since they don't look grossly abnormal in comparison to the naïve mice. There doesn't look to be much airway wall thickening, which is a great indicator of airway remodeling.

Reply#15: We acknowledge that the image shown for airway smooth muscle in mice with established chronic experimental asthma is not representative (Figure 1C, second from left), and was included erroneously. We have now repeated the IHC for all of the groups and included representative images. We have also now included images for collagen deposition, see Figure S1A. It is possible that compared to other models of chronic experimental asthma, where investigators typically administer high doses of house dust mite multiple times per week and for prolonged periods (i.e. 5-10 weeks) that the degree of remodelling in our model is not as dramatic. However, the remodelling that develops in our model is highly reproducible and persistent (both the elevated ASM area and IL-33 production)^{8 19}, and in our view, our model more faithfully captures the epidemiology and therapeutic efficacy (e.g. our findings here, and recent

findings with (i) anti-IL-33,⁸ (ii) P2RY13-mediated type 2 inflammation,¹⁹ and (iii) linkage of IL-33 and NETosis,²⁰ all validated using clinical samples).

Minor points for consideration:

3. Check Figure 6C, as there is text over the top of the x axis label that obscures the label.

Reply#16: Thank you for pointing this out. We have now corrected the text.

4. Despite the findings of this study concerning the effects of corticosteroids almost negating the effect of the DP2 antagonism, I feel that it will be difficult to find an asthmatic population that wouldn't be on corticosteroids of some sort to actually test whether the DP2 antagonism would be more effective in the absence of steroids.

Reply#17: We agree wholeheartedly since corticosteroids are presently the first line treatment as per the GINA guidelines. Importantly, we are not advocating that patients should stop their corticosteroid medication; rather our focus was to try and understand why DP2 antagonist therapy did not meet the primary endpoint in the phase III trials, and how these compounds mediate the resolution of airway remodelling. In the future, it may transpire that patients whose asthma is well controlled by biologics will be advised to stop taking corticosteroids. Where the biologics are known not to reverse airway remodelling, then one could envisage a patient receiving the biologic together with a DP2 antagonist (in the absence of steroid). In the context of COPD, we have evidence (unpublished) that shows that DP2 antagonism is effective in reversing airway remodelling in a preclinical model of smoke-induced emphysema. Although steroids may be prescribed for COPD, they are often ineffective and increase the risk of a bacterial infection, and hence DP2 antagonists, which have an excellent safety record, may be suitable for this indication.

5. It would have been good to also have lung function parameters on the mice to determine whether the interventions used had an effect in the absence of steroids, which would've strengthen the data.

Reply#18: We acknowledge that the absence of lung function data is a limitation of the paper, however, such analyses are not trivial and, in our experience, separate cohorts of mice are needed to those where pathology and immune measures are performed, almost doubling the time (and cost) required to complete such studies. Here, based on our hypothesis, we prioritised the immunological mechanism.

References:

1. Sawyer N, *et al.* Molecular pharmacology of the human prostaglandin D2 receptor, CRTH2. *Br J Pharmacol* **137**, 1163-1172 (2002).
2. Saunders R, *et al.* DP(2) antagonism reduces airway smooth muscle mass in asthma by decreasing eosinophilia and myofibroblast recruitment. *Science translational medicine* **11**, (2019).
3. Gonem S, *et al.* Fevipiprant, a prostaglandin D2 receptor 2 antagonist, in patients with persistent eosinophilic asthma: a single-centre, randomised, double-blind, parallel-group, placebo-controlled trial. *Lancet Respir Med* **4**, 699-707 (2016).
4. Mascia K, Haselkorn T, Deniz YM, Miller DP, Bleecker ER, Borish L. Aspirin sensitivity and severity of asthma: evidence for irreversible airway obstruction in patients with severe or difficult-to-treat asthma. *J Allergy Clin Immunol* **116**, 970-975 (2005).
5. Luskin K, Thakrar H, White A. Nasal Polyposis and Aspirin-Exacerbated Respiratory Disease. *Immunol Allergy Clin North Am* **40**, 329-343 (2020).
6. Sanak M, Gielicz A, Bochenek G, Kaszuba M, Nizankowska-Mogilnicka E, Szczeklik A. Targeted eicosanoid lipidomics of exhaled breath condensate provide a distinct pattern in the aspirin-intolerant asthma phenotype. *J Allergy Clin Immunol* **127**, 1141-1147.e1142 (2011).
7. Toussaint M, *et al.* Host DNA released by NETosis promotes rhinovirus-induced type-2 allergic asthma exacerbation. *Nat Med* **23**, 681-691 (2017).
8. Werder RB, *et al.* Chronic IL-33 expression predisposes to virus-induced asthma exacerbations by increasing type 2 inflammation and dampening antiviral immunity. *J Allergy Clin Immunol* **141**, 1607-1619.e1609 (2018).
9. Loh Z, *et al.* HMGB1 amplifies ILC2-induced type-2 inflammation and airway smooth muscle remodelling. *PLoS Pathog* **16**, e1008651 (2020).
10. Lynch JP, *et al.* Plasmacytoid dendritic cells protect from viral bronchiolitis and asthma through semaphorin 4a-mediated T reg expansion. *The Journal of experimental medicine* **215**, 537-557 (2018).
11. Lynch JP, *et al.* Aeroallergen-induced IL-33 predisposes to respiratory virus-induced asthma by dampening antiviral immunity. *J Allergy Clin Immunol* **138**, 1326-1337 (2016).
12. Sikder MAA, *et al.* Maternal diet modulates the infant microbiome and intestinal Flt3L necessary for dendritic cell development and immunity to respiratory infection. *Immunity*, (2023).

13. Werder RB, *et al.* PGD2/DP2 receptor activation promotes severe viral bronchiolitis by suppressing IFN- λ production. *Science translational medicine* **10**, (2018).
14. Brightling CE, *et al.* Effectiveness of fevipiprant in reducing exacerbations in patients with severe asthma (LUSTER-1 and LUSTER-2): two phase 3 randomised controlled trials. *Lancet Respir Med* **9**, 43-56 (2021).
15. Pettipher R, *et al.* Pharmacologic profile of OC000459, a potent, selective, and orally active D prostanoid receptor 2 antagonist that inhibits mast cell-dependent activation of T helper 2 lymphocytes and eosinophils. *J Pharmacol Exp Ther* **340**, 473-482 (2012).
16. Redington AE, *et al.* Increased expression of inducible nitric oxide synthase and cyclooxygenase-2 in the airway epithelium of asthmatic subjects and regulation by corticosteroid treatment. *Thorax* **56**, 351-357 (2001).
17. Aksoy MO, Li X, Borenstein M, Yi Y, Kelsen SG. Effects of topical corticosteroids on inflammatory mediator-induced eicosanoid release by human airway epithelial cells. *J Allergy Clin Immunol* **103**, 1081-1091 (1999).
18. O'Banion MK, Winn VD, Young DA. cDNA cloning and functional activity of a glucocorticoid-regulated inflammatory cyclooxygenase. *Proc Natl Acad Sci U S A* **89**, 4888-4892 (1992).
19. Werder RB, *et al.* Targeting the P2Y(13) Receptor Suppresses IL-33 and HMGB1 Release and Ameliorates Experimental Asthma. *Am J Respir Crit Care Med* **205**, 300-312 (2022).
20. Curren B, *et al.* IL-33-induced neutrophilic inflammation and NETosis underlie rhinovirus-triggered exacerbations of asthma. *Mucosal Immunol*, (2023).